

# Weather Type Reconstruction using Machine Learning Approaches

Lucas Pfister[1,2], Lena Wilhelm[1,2], Yuri Brugnara*[1,2], Noemi Imfeld[1,2], and Stefan Brönnimann [1,2]

[1]Oeschger Centre for Climate Change Research, University of Bern, Bern, 3012, Switzerland
[2]Institute of Geography, University of Bern, Bern, 3012, Switzerland
*now at Empa, Dübendorf, 8600, Switzerland

**Correspondence:** Lucas Pfister (lucas.pfister@unibe.ch)

**Abstract.** Weather types are used to characterise large–scale synoptic weather patterns over a region. Long–standing records of weather types hold important information about day–to–day variability and changes of atmospheric circulation and the associated effects on the surface. However, most weather type reconstructions are restricted in their temporal extent as well as in the accuracy of the used methods. In our study, we assess various machine learning approaches for station–based weather
type reconstruction over Europe based on the CAP9 weather type classification. With a common feedforward neural network performing best in this model comparison, we reconstruct a daily CAP9 weather type series back to 1728. The new reconstructions constitute the longest daily weather type series available. A detailed validation shows considerably better performance compared to previous statistical approaches and good agreement with the reference series for various climatological analyses. Our approach may serve as a guide for other weather type classifications.

## 1 Introduction

Weather type (WT) or circulation type classifications are a widespread tool to characterize the prevailing large–scale synoptic weather patterns over a specific region (Philipp et al., 2010). In Europe, where daily weather is mainly governed by transient high and low pressure systems driven by the westerly jet stream, such classifications prove particularly useful to describe the prevailing atmospheric conditions. WT time series yield important information about variability and changes of atmospheric
patterns (Jones et al., 2014; Rohrer et al., 2017; Kučerová et al., 2017) and the surface effects associated with them (Paegle, 1974; O'Hare and Sweeney, 1993; Kostopoulou and Jones, 2007; Lorenzo et al., 2008; Jones and Lister, 2009; Casado et al., 2010; Küttel et al., 2011). Various studies have assessed the links between WTs and extreme events such as droughts (Fleig et al., 2010), temperature extremes (Hoy et al., 2020; Sýkorová and Huth, 2020) or extreme precipitation and floods (Minářová et al., 2017; Petrow et al., 2009). Moreover, WT classifications are applied for evaluating weather forecasting model simulations
(Stryhal and Huth, 2019; Weusthoff, 2011) or forecasting in the renewable energy sector (Wang et al., 2022; Drücke et al., 2021; Li et al., 2020), among others.

The first WT classifications were created by experienced meteorologists who classified the atmospheric situation employing manually drawn weather charts derived from station observations (Hess and Brezowsky, 1952; Lamb, 1972; Schüepp, 1979). While these subjective classifications represent real synoptic features, they are often subject to inconsistencies and ambiguities
(e.g. James, 2007; Cahynová and Huth, 2009; Jones et al., 2014; Wanner et al., 2000). In more recent decades, hybrid (mixed) or





objective (automatized) WT classifications have been introduced, that classify atmospheric patterns numerically using various statistical approaches, such as clustering algorithms, class attribution based on a distance measure, or even machine learning approaches (Huth et al., 2008; Mittermeier et al., 2022). Such automatized WT classification is usually based on gridded meteorological data (Huth et al., 2008). Because the temporal coverage of such gridded datasets is limited, WT classifications
usually only reach back several decades. In order to study long–term changes (i.e. over multiple decades or even centuries) in atmospheric circulation patterns and associated surface effects, long–term time series of WT classifications are needed.

With the newest generation of reanalysis datasets, many WT records could already be extended back to the 19[th] century (Philipp et al., 2010; Jones et al., 2014). Currently, the limit for WT classifications based on atmospheric fields is set by the 20[th] Century Reanalysis version 3 (20CRv3; Slivinski et al., 2019; Compo et al., 2011), which extends back to 1806. Prior
to that, historical station observations and qualitative descriptions of the atmospheric conditions from weather diaries are the only sources available for classifying WTs. Recent data rescue and digitisation efforts (Brunet and Jones, 2011; Brönnimann et al., 2019; Pfister et al., 2019; Brugnara et al., 2019, 2020b, 2022b) brought to light a vast amount of early instrumental meteorological records which can be used for this purpose, particularly in central Europe. Only a small number of studies have used this data so far, resulting in some long–term, station–based WT reconstructions starting in the middle of the 18[th] century
(Schwander et al., 2017; Delaygue et al., 2019). However, the main limitation of the station–based reconstructions that are currently available is that they use relatively simple statistical approaches (i.e. the shortest Mahalanobis distance (SMD) from a defined centroid) that only capture the most prominent features of atmospheric circulation patterns and that they are restricted to using continuous data such as pressure and temperature. Especially during the early instrumental period, such quantitative data is scarce, whereas qualitative meteorological information from weather diaries is more widely available. More complex
approaches that can detect patterns in more detail and make use of qualitative data could improve existing WT reconstructions and might even allow for extending them backwards in time, where even less quantitative information is available.

Whereas common statistical approaches seem to have reached their limit for this purpose, supervised machine learning (ML) classification methods present a promising approach in this regard, as they are suited for recognising complex non–linear patterns, which pertain to the distribution of atmospheric variables. Furthermore, they can handle mixed data types, i.e. they
could also include qualitative data on past weather in a categorised form. Nowadays, artificial intelligence is commonly used for classification and pattern recognition in meteorological and climatological research, ranging from detection of extreme events (Racah et al., 2017; Chattopadhyay et al., 2020), frontal systems (Dagon et al., 2022; Bochenek et al., 2021; Biard and Kunkel, 2019), blocking situations (Muszynski et al., 2021; Thomas et al., 2021) or storms and cyclone tracks (Accarino et al., 2023; Kumler-Bonfanti et al., 2020; Mittermeier et al., 2019; Williams et al., 2008). In the context of WT classifications, ML is still
a rather novel approach. Schlef et al. (2019) used neural networks to detect circulation patterns associated with extreme floods in the US. Luferov and Fedotova (2020) used a convolutional neural network to reconstruct Dzerdzeevskii weather types for the northern hemisphere (Dzerdzeevskii, 1962). Mittermeier et al. (2022) studied WT pattern changes in the context of climate change using ML classifications of the Grosswetterlagen (general weather types) for central Europe after Hess and Brezowsky (1952). Whereas this pioneering work of WT reconstruction is entirely based on gridded data from atmospheric reanalyses,





an application of ML approaches to station–based WT classification in order to reconstruct long–term WT series is currently
      lacking.

      In our study, we address this gap by assessing different machine learning approaches for station–based WT reconstruction
      over Europe. For this method intercomparison, we use the CAP7 WT classification created by Schwander et al. (2017), which
      is a simplification of the CAP9 WT classification representative of central Europe (Weusthoff, 2011). As CAP7 is an objective
WT classification based on a cluster analysis of principal components from reanalysis pressure data, it does not suffer from
      the aforementioned issues with subjective WT classes and thus provides an ideal testbed for training and evaluating our ML
      approaches. Our study pursues two aims: i) providing an encompassing assessment of different ML approaches for the pur-
      pose of objective WT classification using station observations and ii) extending the CAP9 WT reconstruction to the period
      1728–2022. Our assessment of the ML approaches is performed with the same input data that Schwander et al. (2017) used for
their Mahalanobis distance–based approach, which serves as a baseline for comparison. We assess logistic regression, random
      forests, as well as classical, recurrent and convolutional neural network approaches. The most powerful model from this com-
      parison is then retained to reconstruct daily CAP9 weather types back to 1728 from an extended set of station data. For this
      reconstruction, additional station series that became available in recent years were included (see Sect. 2.2). The reliability of
      the WT reconstructions is evaluated in detail to provide a robust basis for eventual applications of this WT series, as well as to
explore possible room for improvement for future attempts in WT classification. Furthermore, we provide a short assessment
      of the impact of including time series of wet days as model input. A more encompassing analysis of the effect of using qualita-
      tive data for WT reconstruction – especially data on wind direction which would provide valuable information on atmospheric
      circulation – must be left for future research as currently long–term, homogeneous time series are virtually inexistent.

      The article is organised as follows: Sect. 2 gives an overview of the data and machine learning approaches used for WT
reconstruction, as well as the model tuning strategy. Results and discussion are presented in Sect. 3. The first part shows a
      detailed intercomparison of the station–based WT reconstruction methods on the example of CAP7 WTs. The second part
      analyses the extended CAP9 reconstruction using the best model from the comparison. Summary and conclusions are given in
      Sect. 4.

## 2  Data & Methods

### 2.1  Weather types

From the abundant number of WT classifications for Europe (see Philipp et al., 2010, 2016, for an overview), we use the
CAP9 WT classification as produced and continuously updated by MeteoSwiss (Weusthoff, 2011). The CAP9 classification
was chosen as it is objective (see discussion in Sect. 1) and because it has been shown to be a reliable predictor of surface
climatic conditions in the Alpine region (Schiemann and Frei, 2010). Furthermore, a manageable amount of nine WTs – e.g.
compared to the 29 WTs after Hess and Brezowsky (1952) – was found to be more suitable for assessing our ML approaches.
Given the scarcity of meteorological records in the early instrumental period, classifications with abundant WTs could not be
accurately represented by the few observation sites.



This WT classification is based on the CAP (Cluster Analysis of Principal Components) method (for details see Weusthoff, 2011; Philipp et al., 2010; Comrie, 1996; Ekström et al., 2002): in a first step, atmospheric variables are decomposed into

their principal components. The principal component time series are then clustered in a second step to derive WT classes. The CAP9 classification by MeteoSwiss was derived from mean sea level pressure from the ERA40 reanalysis (Kållberg et al., 2004; Uppala et al., 2005), whereas the attribution to the nine WTs in operational use is based on the Euclidean distance from the respective pressure centroids of the ERA40–derived WTs (Weusthoff, 2011).

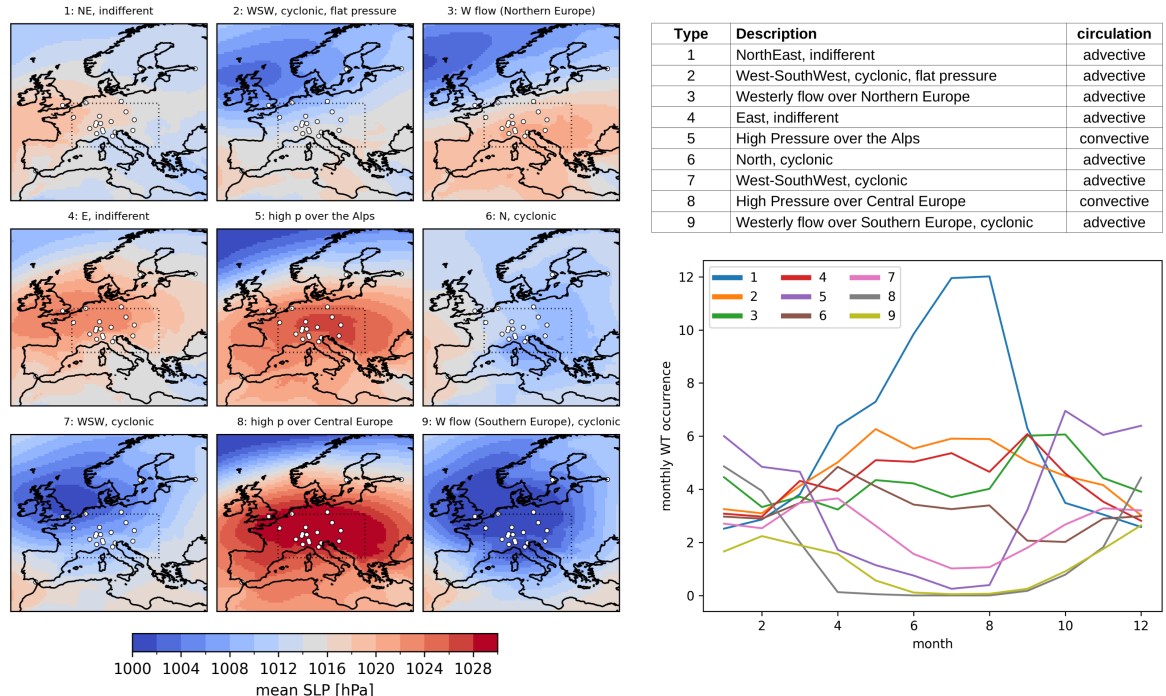

**Figure 1.** Left: climatological average of sea level pressure 1957–2020 for CAP9 weather types. White filled circles indicate station locations (see Sect. 2.2). The dotted rectangle represents the wider Alpine area for which the CAP9 WT classification is representative. Right: description of CAP9 WTs (top) and their average monthly occurrence 1957–2020 (bottom).

The daily time series of CAP9 weather types from 01.09.1957–31.12.2020 used as predictand for the model training and
as reference series for the analyses in Sect. 3 was obtained from MeteoSwiss. An overview of the synoptic situation of the different WTs is given in Fig. 1 (left). Shown are filled contours of average sea level pressure derived from the ERA5 reanalysis (Hersbach et al., 2020; Bell et al., 2021) over the period 1957–2020. Whereas there are seven types associated with advective patterns for the Alpine region, only WTs 5 and 8 are dominated by convective circulation (Fig. 1, top right; see also Weusthoff, 2011). Note that the CAP9 WTs have different occurrence frequencies with some showing strong seasonal patterns (Fig. 1,
bottom right). For our model comparison (Sect. 3.1), we use a reduced set of seven weather types (CAP7) in order to compare the results directly with the Mahalanobis distance approach in Schwander et al. (2017). They found types 5 and 8, as well as 7



and 9 in the CAP9 classification hard to distinguish and merged the respective WT pairs. While we merge the same pairs for the analyses in Sect. 3.1, the machine learning models are trained on the original CAP9 WTs.

## 2.2 Station observations

Meteorological observations used for WT reconstruction are located around and within the greater Alpine region in central Europe, for which the CAP9 classification is representative (Fig. 1; see also Weusthoff, 2011). Note that the available stations are well distributed across most parts of Europe, which is crucial to capture the large–scale synoptic situation. However, in southern and eastern Europe available digitised station records unfortunately were scarce. Whereas the CAP9 classification is based solely on pressure data, the station observations used for our reconstructions also include other variables, i.e. temper-
ature and categorical rain data. Pressure data represents the synoptic atmospheric flow, whereas the other variables represent the associated surface effects and thus may provide valuable additional information for WT reconstruction (Schwander et al., 2017), especially in the context of the early instrumental period with scarce data availability. A summary of the available daily station records is given in Table 1 with the data source indicated in the last column.

Table 1: daily meteorological data used for WT reconstructions. t = temperature, p = pressure, $\Delta p$ = temporal pressure gradient, rr = wet days

| ID | Name | Lat | Lon | Alt (m a.s.l.) | Variables | Period | Source / Comments |
|---|---|---|---|---|---|---|---|
| BAS | Basel | 47.541 | 7.584 | 316 | t, | 1756–2020 | CHIMES (Brönnimann and Brugnara, 2020, 2021), |
| | | | | | p, $\Delta p$ | 1764–2020 | MeteoSwiss (Füllemann et al., 2011; Begert et al., |
| | | | | | rr | 1864–2020 | 2005) |
| BER | Bern | 46.991 | 7.464 | 552 | t, | 1781–2020 | CHIMES (Brugnara et al., 2022a), MeteoSwiss |
| | | | | | p, $\Delta p$ | 1781–2020 | (Füllemann et al., 2011; Begert et al., 2005) |
| BRL | Berlin | 52.456 | 13.300 | 40 | p, $\Delta p$ | 1728–2020 | DWD (Behrendt et al., 2011; Kaspar et al., 2013); |
| | | | | | rr | 1876–2020 | gap in pressure series 1771–1875 |
| BOL | Bologna | 44.497 | 11.353 | 53 | t | 1728–2020 | Camuffo et al. (2017), ECA&D (Klein Tank et al., |
| | | | | | rr | 1818–2020 | 2002) |
| CAD | Cadiz | 36.500 | –6.260 | 1 | t | 1790–2020 | IMPROVE (Camuffo and Jones, 2002; Barriendos |
| | | | | | p, $\Delta p$ | 1818–2020 | et al., 2002), ECA&D (Klein Tank et al., 2002) |
| DBL | DeBilt | 52.100 | 5.180 | 1 | t | 1738–2020 | ECA&D (Klein Tank et al., 2002), Brandsma et al. |
| | | | | | p, $\Delta p$ | 1738–2020 | (2000) |
| ENG | Engelberg | 46.822 | 8.411 | 1035 | rr | 1864–2020 | MeteoSwiss (Füllemann et al., 2011; Begert et al., 2005) |
| GVA | Geneva | 46.248 | 6.128 | 410 | t | 1771–2020 | CHIMES/DigiHom (Häderli et al., 2020; |
| | | | | | p, $\Delta p$ | 1818–2020 | Brönnimann et al., 2020), MeteoSwiss (Füllemann |
| | | | | | rr | 1864–2020 | et al., 2011; Begert et al., 2005) |
| HPE | Hohenpeissenberg | 47.800 | 11.020 | 995 | t | 1781–2020 | Winkler (2006, 2009), DWD (Behrendt et al., 2011; Kaspar et al., 2013) |
| | | | | | p, $\Delta p$ | 1781–2020 | |
| | | | | | rr | 1818–2020 | |
| KAR | Karlsruhe | 49.039 | 8.365 | 112 | t | 1764–2020 | Brugnara et al. (2015), DWD (Behrendt et al., 2011; Kaspar et al., 2013), ECA&D (Klein Tank et al., 2002); gaps 1790–1818, 1864–1876 |



| LOH | Lohn | 47.752 | 8.678 | 585 | rr | 1864–2020 | MeteoSwiss (Füllemann et al., 2011; Begert et al., 2005) |
| LDN | London | 51.515 | –0.120 | 1035 | p, Δ p | 1728–2020 | Cornes et al. (2012a), ECA&D (Klein Tank et al., 2002) |
| LUG | Lugano | 46.000 | 8.970 | 273 | t | 1864–2020 | MeteoSwiss (Füllemann et al., 2011; Begert et al., 2005) |
| | | | | | p, Δ p | 1864–2020 | |
| | | | | | rr | 1864–2020 | |
| MIL | Milan | 45.470 | 9.180 | 132 | t | 1764–2020 | IMPROVE (Moberg et al., 2000; Maugeri et al., 2002), ECA&D (Klein Tank et al., 2002) |
| | | | | | p, Δ p | 1874–2020 | |
| OXF | Oxford | 51.760 | –1.260 | 63 | rr | 1864–2020 | ECA&D (Klein Tank et al., 2002) |
| PAD | Padua | 45.398 | 11.800 | 12 | t | 1781–2020 | IMPROVE (Camuffo and Jones, 2002; Camuffo et al., 2006), Brugnara et al. (2015), ECA&D (Klein Tank et al., 2002) |
| | | | | | p, Δ p | 1728–2020 | |
| PAR | Paris | 48.817 | 2.322 | 77 | t | 1876–2020 | Cornes et al. (2012b), ECA&D (Klein Tank et al., 2002) |
| | | | | | p, Δ p | 1749–2020 | |
| PRA | Prague | 50.090 | 14.420 | 190 | t | 1781–2020 | Kyselý (2007), Stepanek (2005), ECA&D (Klein Tank et al., 2002) |
| SAM | Samedan | 46.526 | 9.879 | 1708 | rr | 1864–2020 | MeteoSwiss (Füllemann et al., 2011; Begert et al., 2005) |
| STK | Stockholm | 59.350 | 18.050 | 44 | t | 1756–2020 | IMPROVE (Moberg et al., 2000), ECA&D (Klein Tank et al., 2002) |
| | | | | | p, Δ p | 1756–2020 | |
| | | | | | rr | 1864–2020 | |
| SPE | St. Petersburg | 59.967 | 30.300 | 3 | t | 1756–2020 | IMPROVE (Camuffo and Jones, 2002), ECA&D (Klein Tank et al., 2002) |
| TOR | Turin | 45.070 | 7.680 | 281 | t | 1756–2020 | Di Napoli and Mercalli (2008), ECA&D (Klein Tank et al., 2002) |
| | | | | | p, Δ p | 1818–2020 | |
| UPP | Uppsala | 59.861 | 17.641 | 15 | t | 1728–2020 | IMPROVE (Moberg et al., 2000; Bergström and Moberg, 2002), ECA&D (Klein Tank et al., 2002) |
| | | | | | p, Δ p | 1728–2020 | |
| WIE | Vienna | 48.249 | 16.356 | 198 | t | 1781–2020 | GeoSphere Austria (2021); gap in temperature series 1818–1864 |
| | | | | | p, Δ p | 1864–2020 | |
| | | | | | rr | 1864–2020 | |
| ZAG | Zagreb | 45.820 | 15.980 | 156 | t | 1864–2020 | ECA&D (Klein Tank et al., 2002) |
| | | | | | p, Δ p | 1864–2020 | |
| SMA | Zurich | 47.378 | 8.566 | 555 | t | 1764–2020 | CHIMES (Brugnara et al., 2022a), MeteoSwiss (Füllemann et al., 2011; Begert et al., 2005); gap in pressure series 1790–1818 |
| | | | | | p, Δ p | 1764–2020 | |
| | | | | | rr | 1864–2020 | |

For the comparison of reconstruction methods (Sect. 3.1), we use the same set of stations and variables that were used by Schwander et al. (2017) without any further preprocessing (see the SMD station sets in Fig. 2). This encompasses station records from London (Cornes et al., 2012a), Milan, Uppsala, Stockholm (Moberg et al., 2000; Maugeri et al., 2002), Turin



(Di Napoli and Mercalli, 2008), Prague (Kyselý, 2007; Stepanek, 2005; Brázdil et al., 2012), Hohenpeissenberg (Winkler, 2009), De Bilt (Klein Tank et al., 2002), Paris (Cornes et al., 2012b, only temperature), Bern, and Lugano (Füllemann et al.,

2011; Begert et al., 2005). Using the same data allows for a direct comparison between our machine learning approaches and the Mahalanobis distance–based method used in Schwander et al. (2017). In accordance with the latter study, daily mean temperature, sea level pressure and the computed pressure difference to the previous day were used as input variables for this comparison.

Further early instrumental station series have been made available as a result of data rescue efforts in recent years (Brön-
nimann et al., 2019; Brugnara et al., 2020b), enhancing the data coverage in our area of interest and extending the period for which WTs can be reconstructed. Unfortunately, the majority of available records covers only a few years and thus is not suitable for our purpose. Using short observation records would lead to varying sets of stations, which on the one hand would introduce inconsistencies in reconstructed WTs and on the other hand constitute immense computational efforts, as for each set of stations a new model has to be trained. Further issues arise from inhomogeneities in the observation series in time (e.g.
observation errors, artificial trends or shifts), which originate from changes in instruments or observation sites, as well as various error sources related to early instrumental data (see e.g. Brugnara et al., 2020a; Winkler, 2006; Böhm et al., 2010). Such inhomogeneities would again lead to errors or biases in the reconstructed WT series.

Where possible, long–term, homogenised station records that contain no or only few and short gaps were used for our approach. For some locations, however, multiple historical observation records from the same location had to be merged into a
single time series. For the temperature series from Bern, Basel, Geneva and Zurich, we could benefit from previous efforts to merge and homogenise daily temperature series (Brugnara et al., 2022a). Only stations at close locations, i.e. within a radius of less than 15 km, have been merged, with the exceptions of Cadiz (merged with T and p data from Huelva) and De Bilt (merged with T data from Haarlem and p data from Zwanenburg, Haarlem, Den Helder and Delft), where the existing series could not be complemented with nearby station records. Complementary series have been retrieved from the ECA&D database (Klein Tank
et al., 2002), as well as from the databases of MeteoSwiss (Füllemann et al., 2011; Begert et al., 2005), the German weather service DWD (Behrendt et al., 2011; Kaspar et al., 2013), the Royal Netherlands Meteorological Institute KNMI (Brandsma et al., 2000), and (GeoSphere Austria, 2021, formerly Austrian Central Institution for Meteorology and Geodynamics ZAMG). The station sets used for the reconstruction of CAP9 WTs (Sect. 3.3.1) are summarised in Fig. 2 labeled according to their respective start date.

Whereas in Schwander et al. (2017) observation records had not been homogenised, we deemed it suitable to apply such a procedure to all pressure and temperature series that had not been homogenised, as well as to the merged series. We used the break point detection approach by Wang and Feng (2018) combining a penalised maximal t test (Wang et al., 2007) and a penalised maximal F test (Wang, 2008). As reference series, we used monthly pressure and temperature series extracted for the respective station locations from the EKF400v2 reanalysis (Valler et al., 2022). For further details on this homogenisation
approach, see Imfeld et al. (2023). Most of the homogenised station records exhibit no or smaller gaps with a median of 31 days. All gaps up to a length of 5 years were imputed with a k nearest neighbor approach following Batista and Monard (2002). This is the same approach also used by Schwander et al. (2017) for their WT reconstructions, thus keeping the consistency



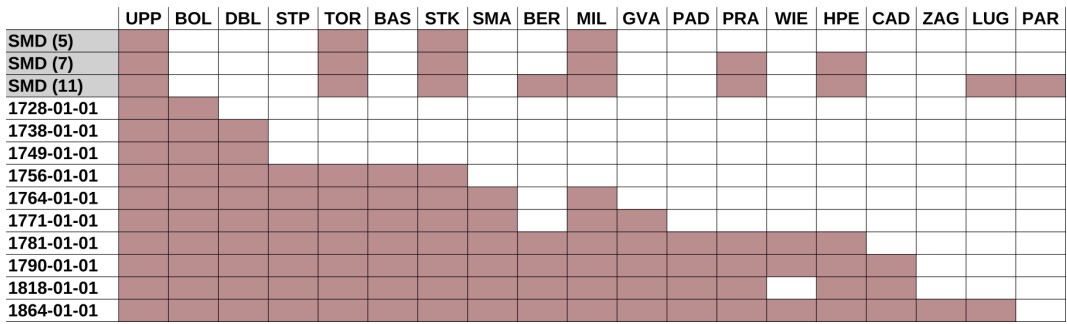

**Figure 2.** Station sets of a) pressure and b) temperature used for the model comparison. The top three rows (SMD, grey shaded) refer to the station sets in Schwander et al. (2017) with 5, 7 and 11 stations, respectively. Station sets indicated with a date are used for the CAP9 reconstruction. The date refers to the start date of the respective station set. Data availability is indicated by the filled blue (pressure) and red (temperature) squares.

in our datasets. Tests for the imputation approach with 25 % randomly introduced gaps revealed an average bias of –0.063 hPa (–0.05 °C) and a mean absolute error of 1.83 hPa (1.46 °C) for pressure (and temperature). We thus deemed this method suitable for the task of WT reconstruction. The series from Berlin, Karlsruhe, Vienna (temperature) and Zurich (pressure) have longer gaps in their station record, which were kept.

Further preprocessing was necessary to use the station observations in the different machine learning models (the results of the respective assessments are not shown). First of all, a global warming trend is visible in all temperature records. In order to establish robust classification models, such non–stationarities in the data had to be removed. Temperature trends were removed individually for each series using a 3$^{\text{rd}}$ order polynomial fit. Furthermore, the pronounced seasonality of temperature might blur temperature signals originating from atmospheric dynamics and lead to an inhomogeneous treatment of weather types throughout the year. Thus, temperature data were corrected for their seasonality by fitting the first two harmonics to each temperature record, which was then subtracted from the data. Pressure and precipitation data have not been corrected for a trend or seasonality, as their contribution to the total variability of these variables was assumed negligible. All variables





from all stations were standardised (i.e. by subtracting their average and dividing by their standard deviation). As pressure gradients and thus atmospheric patterns are less pronounced in summer than in winter (see e.g. Fig. 5 in Sect. 3.3), a monthly standardisation of pressure was tested (not shown). However, this deteriorated the reconstructions and was thus dismissed.

## 2.3   Machine Learning Approaches

For our model comparison (Sect. 3.1) multiple machine learning models are tested and compared against a baseline WT

classification approach. This baseline model is given by the simple statistical classification approach by Schwander et al. (2017) for their CAP7 reconstructions and is based on the shortest Mahalanobis distance (SMD) of station observations to the centroids (station data averages) for each WT previously calculated from the reference period data. Further details on this approach are expounded in Schwander et al. (2017). The focus of this section lies on the ML approaches, including a multinomial logistic regression model, a random forest model, feed forward neural networks, as well as recurrent and convolutional neural networks.

The best performing model is then selected for the reconstruction of daily CAP9 WTs back to 1728 (see Sect. 3.3).

### 2.3.1   Multinomial Logistic Regression (MLG)

Multiple logistic regression is a commonly used method for classification problems with categorical outcome. With a multiple logistic regression model, we can predict the occurrence probability $p$ of a weather class $WT$ as a function of several different station observations $x_1, x_2, ..., x_n$ as independent variables (Hosmer and Lemeshow, 2000). Whereas multiple logistic regres-

sion can predict only a binary dependent variable $y$, multnomial logistic regression can handle several response classes (given that they have no natural order). The occurrence probability $p(x)$ is defined as:

$$y = p(x) = \frac{1}{1 + e^{(-g(x))}}, \text{ where } 0 \le p(x) \le 1$$

The model is based on a linear regression function $g(x)$:

$$g(x) = \beta_0 + \beta_1 x_1 + \beta_2 x_2 + ... + \beta_n x_n$$

The regression coefficients $\beta_n$ are computed applying the maximum likelihood method to maximise the probability, meaning that the coefficients are determined iteratively. For details see the documentation of R's caret package (Kuhn, 2008).

Compared to complex and more advanced machine learning methods, logistic regression has the advantage of interpretability, as the relationships between predictors and predictand can be directly inferred. One major drawback, however, is that often only a small number of covariates can be used in a model, as an increasing number of covariates may be subject to multi-collinearity and can lead to overfitting of the model. To avoid this, we limited the number of predictors to five and constrained the variance inflation factor (VIF) to values below four. Furthermore, one has to keep in mind that logistic regression only al-

lows for a linear combination of covariates, thus non–linear features in the predictor data with respect to WTs are not captured by MLG.





### 2.3.2 Random Forests (RF)

The second machine learning approach assessed in this paper are random forests (RF) (Ho, 1995; Breiman, 2001). In contrast to single decision trees, RF use an ensemble of decision trees built from subsamples of the training data. With an increasing number of trees, the generalisation error of RF models decreases and robust predictions can be established. In the case of our classification application, RF can provide a probabilistic estimate of the true WT using their ensemble of decision trees. Compared to other machine learning approaches, RF are fast to train (depending on the number of trees), but can suffer from overfitting. In order to find a RF architecture with an optimal balance between accuracy and generalisability, several parameter sets are tested. These encompass the number of trees (between 10 and 400), the maximum depth (between 5 and 30), the minimum sample size for splitting (between 2 and 10) and the minimum sample size for a leaf (between 1 and 4). Furthermore, the Gini impurity and entropy were tested for determining the splits. For further information see the documentation of the scikit–learn python package (Pedregosa et al., 2011).

### 2.3.3 Feedforward Neural Networks (NN)

The second approach are feedforward neural networks (NN) (Rosenblatt, 1958; Hastie et al., 2009). Similar to the RF approach, NN provide estimates of probability for each class, represented by the normalised weights of the output layer. The NN architecture used for our work is not based on a pre–designed NN model. While we prescribed the use of multiple layers, including a dropout layer before the output layer to avoid overfitting, optimal architectural properties such as the number of layers and their sizes were determined from scratch with a hyperparameter search on the training data (see also Sect. 2.4). In particular, networks with a number of layers between 2 and 8 were tested with layer sizes between 32 and 256 (in steps of 32). Furthermore, dropout rates between 0.05 and 0.2 (in steps of 0.05), as well as learning rates between $10^{-4}$ and $10^{-2}$ were tested during model tuning. The models were trained using the Adam optimisation algorithm (Kingma and Ba, 2014) and the categorical crossentropy loss function. We set the batch size to 200 and the maximum number of epochs to 50 (with early stopping with a patience of five epochs). The NN approach, as well as the other neural network approaches were implemented using Tensorflow (Abadi et al., 2016a, b) and Keras (Chollet, 2021) libraries.

### 2.3.4 Recurrent and Convolutional Neural Networks (RNN & CNN)

Both, the RF and the NN models described above use input data from the same day as predictors. As circulation patterns can persist for several days, it might be beneficial to also include information from preceeding days in our models. For this reason, we assess both recurrent neural networks (RNN) and 1D–convolutional neural networks (CNN) in this study. For the RNN we used so–called long–short–term memory networks (LSTM) that can retain or discard information from previous time steps, thus being able to propagate relevant information over multiple time steps (Hochreiter and Schmidhuber, 1997). Our RNN architecture follows the one of the NN, again with a dropout layer before the output layer and the same settings for model training. For reasons of computational costs, less architectural configurations were assessed than for the NN (i.e. between 2 and 5 layers with sizes between 32 and 128).





Similar to RNNs, convolutional neural networks (CNN) can also make use of data from previous timesteps. Whereas CNN
is mostly applied to image data or other multi–dimensional datasets for pattern detection using trained filters (Fukushima,
1980), we used its 1–dimensional equivalent for time series analysis (Kiranyaz et al., 2021). Like for the RNN, a reduced set
of architectural properties (i.e. between 2 and 5 layers with sizes between 32 and 128) has been assessed, while the rest of the
tunable parameters were kept identical to the other networks.

For both time–dependent neural networks (RNN and CNN), we used data from two days prior to the day of interest (three
days in total) to predict the WTs. A longer time window was found not to yield improvements in the results (not shown).
Analogous to NN, RNN and CNN were also trained using the Adam optimisation algorithm with the categorical crossentropy
loss function, a batch size of 200 and a maximum of 50 epochs with early stopping.

## 2.4  Hyperparameter Tuning and Validation

Training and validation of the machine learning approaches was performed with the data described in Sect. 2.1 and 2.2 using
the station observations as predictors and the CAP9 WT classification as predictand. After preliminary tests with certain sub-
sets of stations and atmospheric variables (not shown), which did not yield any clear gain in performance, we chose to use all
available pressure and temperature series. For their approach, Schwander et al. (2017) used a reduced set of seven WTs (CAP7).
Two pairs of WTs, 5 (high pressure over the Alps) and 8 (high pressure over central Europe), as well as 7 (west–southwest,
cyclonic) and 9 (westerly flow over southern Europe, cyclonic) were combined to single WTs, as they were found to be too
similar to distinguish. In order to compare our machine learning models to the SMD approach in Schwander et al. (2017) in
our model comparison, they are trained on the same station data as used in the original study, but with the CAP9 WTs as pre-
dictand. To make validation measures comparable to the baseline model, CAP9 classes are subsequently converted to CAP7 by
combining the pairs of WTs accordingly. Also the reference period for the model comparison (Sect. 3.1) was chosen similar to
the baseline study by Schwander et al. (2017), spanning 01.01.1961–31.12.1998. For our new WT reconstructions (Sect. 3.3),
we made use of the full available period for model training spanning 01.09.1957–31.12.2020 and used the CAP9 classification
for the evaluation.

Note that the same data is used for both, hyperparameter tuning and validation of the models. In order to ensure indepen-
dence between model tuning and evaluation, a nested cross–validation (Cawley and Talbot, 2010) is implemented. For the RF
and neural network approaches, an outer loop splits the data into a training and an independent test set. An inner loop is applied
to the training set for hyperparameter tuning, again splitting off part of the data for validation of the model configurations in
order to find the optimal hyperparameters independent from the training data. The outer loop then serves to independently
estimate the validation metrics. Optimal hyperparameters are determined using Bayesian optimisation (Snoek et al., 2012). A
total number of 8 folds for the outer loop and 7 folds for the inner loop without shuffling and without overlap are applied. For
the MLG model, we followed the same structure of outer and inner loops, but with 10 outer and 10 inner folds (with overlap)
instead of 8 and 7. The outer loop splits the data randomly into 80 % training and 20 % independent testing datasets. The inner
loop uses the 80 % folds for finding the best combination of station variables, again splitting into 70 % of the data for training



and 30 % for validation. We find the best combination and best number of predictors manually with a bidirectional stepwise approach, looking at mean performance, significance and z values of predictors. Once a model was found that worked well on all 10 inner folds and showed a good balance between over– and underfitting, we retrained it with the 80 % sets and evaluated with the independent test sets (20 %) in the outer loop.

As Schwander et al. (2017) did not perform an independent validation of their approach, validation measures are not comparable. For this reason, we reconstructed their approach and applied a cross–validation with the same training and test splits as in the eight outer loops described above. Results from this independent cross–validation can be directly compared to our approaches. When reconstructing the Mahalanobis distance approach of Schwander et al. (2017), an error in their model setup became apparent: when calculating the distance to each WT centroid using the covariance matrix *derived for the respective* WT, considerably lower accuracies than indicated in the original study were obtained (not shown). However, using the covariance matrix from the *true (observed)* WT, which of course would be unknown for the reconstructions, accuracies reached the values from the original study. For our validation of the SMD approach, the distance was calculated for each WT centroid using the correct covariance matrix of the respective WT.

Model performance is estimated with the overall accuracy and average Heidke skill score (HSS; Heidke, 1926; Cohen, 1960) values for all weather types and all seasons. The overall accuracy represents the fraction or percentage of days for which the WTs were correctly classified. The HSS represents the proportion of correct predictions scaled by the expected correct forecasts due to chance for categorical forecasts (see Hyvärinen, 2014) and is calculated for each WT. In contrast to overall accuracy, the HSS accounts for differences in the occurrence of individual WTs. To obtain a robust and independent estimate of the true performance of the best models, an average of these validation measures is taken over the outer folds of the nested cross–validation (i.e. ten and eight test sets for MLG and the other approaches, respectively). Note that the model used for the WT time series reconstruction is retrained with the full available dataset within the validation period. Indicated accuracies for the individual models are thus arguably pessimistic.

## 3 Results & Discussion

### 3.1 Model Intercomparison for CAP7 weather types

The performance of the WT classification approaches presented in Sect. 2.3, as well as the SMD approach by Schwander et al. (2017) for the CAP7 WT classification is indicated in Table 2. Shown accuracies and HSS represent an average from the k–fold cross–validation over the period 01.01.1961–31.12.1998 (see Sect. 2.4) based on three different subsets with data from five, seven and eleven stations, respectively, as used in (Schwander et al., 2017, see also Table 3 therein). For the logistic regression model, only results from the optimal selection of station series is shown (see Sect. 2.3). This best–performing model uses the following six variables: pressure in Milan and Paris, temperature in Prague and Stockholm, and the temporal pressure gradient in Milan and Stockholm.





Table 2: Average accuracy (acc) in percent and average Heidke skill scores (HSS) of all applied approaches for CAP7 WT reconstruction, as well as the baseline model (SMD, grey shaded) using different data subsets. Shown are values for the whole year (ANN), and the individual seasons (winter: DJF, spring: MAM, summer: JJA, autumn: SON). Highest values per station set are marked in bold

| Station Set | Model | ANN | DJF | MAM | JJA | SON |
|---|---|---|---|---|---|---|
| custom selection of variables & stations | MLG | Acc = 74.5 | Acc = 74.3 | Acc = 74.4 | Acc = 73.8 | Acc = 75.3 |
| | | HSS = 0.70 | HSS = 0.71 | HSS = 0.70 | HSS = 0.67 | HSS = 0.71 |
| SMD (5 stations) | SMD | Acc = 64.7 | Acc = 73.3 | Acc = 62.9 | Acc = 56.3 | Acc = 66.3 |
| | | HSS = 0.58 | HSS = 0.60 | HSS = 0.56 | HSS = 0.45 | HSS = 0.58 |
| | RF | Acc = 74.3 | Acc = 78.4 | Acc = 71.8 | Acc = 72.2 | Acc = 75.1 |
| | | HSS = 0.70 | HSS = 0.70 | HSS = 0.67 | HSS = 0.63 | HSS = 0.69 |
| | NN | Acc = 76.1 | Acc = 79.9 | Acc = 73.7 | Acc = 73.7 | Acc = 77.1 |
| | | HSS = 0.72 | **HSS = 0.72** | HSS = 0.70 | **HSS = 0.65** | HSS = 0.72 |
| | RNN | **Acc = 76.8** | **Acc = 80.6** | **Acc = 75.1** | **Acc = 73.8** | **Acc = 77.9** |
| | | **HSS = 0.73** | **HSS = 0.72** | **HSS = 0.71** | **HSS = 0.65** | **HSS = 0.73** |
| | CNN | Acc = 76.0 | Acc = 79.2 | Acc = 74.8 | Acc = 72.4 | Acc = 77.7 |
| | | HSS = 0.72 | HSS = 0.71 | **HSS = 0.71** | HSS = 0.63 | HSS = 0.72 |
| SMD (7 stations) | SMD | Acc = 67.4 | Acc = 75.7 | Acc = 66.3 | Acc = 59.0 | Acc = 68.8 |
| | | HSS = 0.61 | HSS = 0.64 | HSS = 0.61 | HSS = 0.48 | HSS = 0.61 |
| | RF | Acc = 78.4 | Acc = 80.9 | Acc = 77.6 | Acc = 75.8 | Acc = 79.2 |
| | | HSS = 0.75 | HSS = 0.73 | HSS = 0.74 | HSS = 0.67 | HSS = 0.74 |
| | NN | **Acc = 81.6** | **Acc = 84.5** | **Acc = 81.1** | Acc = 78.3 | **Acc = 82.4** |
| | | **HSS = 0.78** | **HSS = 0.78** | **HSS = 0.78** | HSS = 0.71 | **HSS = 0.78** |
| | RNN | Acc = 80.5 | Acc = 83.1 | Acc = 79.5 | Acc = 78.1 | Acc = 81.3 |
| | | HSS = 0.77 | HSS = 0.76 | HSS = 0.76 | HSS = 0.71 | HSS = 0.77 |
| | CNN | Acc = 81.3 | Acc = 83.3 | Acc = 80.4 | **Acc = 79.4** | Acc = 81.9 |
| | | **HSS = 0.78** | HSS = 0.76 | HSS = 0.77 | **HSS = 0.72** | **HSS = 0.78** |
| SMD (11 stations) | SMD | Acc = 62.9 | Acc = 70.6 | Acc = 61.1 | Acc = 55.1 | Acc = 64.8 |
| | | HSS = 0.56 | HSS = 0.56 | HSS = 0.55 | HSS = 0.44 | HSS = 0.56 |
| | RF | Acc = 82.6 | Acc = 83.6 | Acc = 82.0 | Acc = 81.2 | Acc = 83.7 |
| | | HSS = 0.79 | HSS = 0.77 | HSS = 0.79 | HSS = 0.73 | HSS = 0.80 |
| | NN | **Acc = 85.7** | Acc = 87.8 | **Acc = 84.8** | Acc = 83.8 | **Acc = 86.6** |
| | | **HSS = 0.83** | HSS = 0.82 | **HSS = 0.82** | HSS = 0.78 | **HSS = 0.83** |
| | RNN | Acc = 85.4 | **Acc = 88.2** | Acc = 84.6 | Acc = 83.1 | Acc = 85.8 |
| | | **HSS = 0.83** | **HSS = 0.83** | **HSS = 0.82** | HSS = 0.78 | HSS = 0.82 |
| | CNN | Acc = 85.5 | Acc = 87.2 | Acc = 84.7 | **Acc = 84.4** | Acc = 85.8 |
| | | **HSS = 0.83** | HSS = 0.82 | **HSS = 0.82** | **HSS = 0.79** | HSS = 0.82 |

Evidently, all ML approaches outperform the baseline model (SMD, grey shaded) for all sets of stations. With an independent validation and correcting the error in the SMD model (see Sect. 2.4), accuracies are by far lower than indicated in Schwander et al. (2017) dropping below 70 % overall and below 60 % in the summer months. The machine learning approaches show





accuracies of about 75 % even for the smallest set of stations (and the selection of the MLG). Validation measures improve

with the number of stations, reaching a maximum overall accuracy of 85.7 % for the NN model with 11 stations. Note that in contrast, the SMD approach shows lower accuracy values for the largest station set than for the other two, pointing to issues arising from data quality or the spatial distribution of the station network for this approach. Heidke Skill Scores (HSS) show a similar pattern with scores between 0.7 and 0.83 (compared to values between 0.56 to 0.61 for SMD). The superiority of the machine learning approaches might be explained by their ability to capture details in the data and non–linear effects better than

common statistical approaches (see also Sect. 2.3)

From the seasonal validation measures we see a slight drop in accuracy (stronger for the HSS) for spring and summer, which was also found in Schwander et al. (2017), especially for summer. Weaker pressure gradients hamper a robust detection of weather types for these months. The difference between spring/summer and autumn/winter, however, is much smaller for the machine learning approaches compared to SMD. All of our models are thus better capable of coping with seasonal differences.

Random forests and multinomial logistic regression allow some inference about the stations and variables that prove to be the most crucial for WT classification. Regarding the spatial distribution of the stations, it is less a high density of stations within the area for which the CAP9 classification is representative (see Fig. 1), but rather an even distribution of stations around the borders of this area that lead to the most accurate predictions. This becomes evident for the optimal selection in the MLG

approach with all predictors being highly significant in the model ($p \leq 0.05$). The MLG coefficients for each covariant and for each weather type are listed in the supplement (Sect. S.2), together with further illustrations displaying the relationship of each predictor to the probability of each class response in the model. Also, RF results underpin that a spatially well distributed station network is crucial for a robust WT classification. This is not surprising, as for WT classification the models benefit not from the localised effects in the station observations but rather the overall atmospheric signal seen in a combination of

information. In this context, more stations located in southern and eastern Europe (compare Fig. 1) could improve the accuracy of the models. Looking at the feature importance (i.e. for each feature (predictor) the average reduction of the Gini impurity or entropy in the split classes over all trees) in RF, pressure data show the highest importance, followed by temperature (see Sect. S.3 in the supplement). The temporal pressure gradient on the other hand showed lower importance values by one order of magnitude compared to the other variables. These results are robust also to the MLG model, where pressure showed the high-

est importance, followed by temperature and pressure gradient. We want to note, however, that the MLG models still always preferred a combination of all three types of information instead of using just pressure data. This holds equally for the other approaches where preliminary tests using only pressure data vs. using all variables confirmed the use of our multivariate input data (not shown).

The model comparison revealed the feedforward neural network (NN) to exhibit the highest accuracy and HSS estimates, although on average only slightly better than for RNN and CNN. However, the NN can be considered as the best model for another reason: in contrast to RNN (and a bit less so for CNN), it is considerably faster to train, making it favourable also from the computational resources perspective. Regarding this aspect, it is important to mention that the simplest approaches



we tested (MLG, RF) are much less costly in terms of computation hours than neural networks. Depending on the task and the related goal of accuracy, using these simpler methods is thus highly recommended. From this point on, we will only use the feedforward neural network model for further analyses and the final reconstruction.

## 3.2 The Effect of Categorical Weather Data

As stated in the introduction, ML approaches have the advantage that they can simultaneously process continuous and categorical information. In this section, we assess the effect of including time series of wet days based on rain information (see Sect. 2.2) as additional model input, as they have proven to be very valuable for statistical weather reconstructions (Imfeld et al., 2023). For this purpose we trained an NN model for two different station sets used for our new reconstruction (Sect. 3.3), once without and once with adding the categorical rain series. Model building and validation has again been performed as described in Sect. 2.4. We used the station set available from 1728 (fewest predictors: 4 pressure & 2 temperature series; see Fig. 2) and the one available from 1864 (most predictors: 17 pressure & 18 temperature series; see Fig. 2) to analyse the impact of adding categorical data for different numbers of predictors. Both station sets were complemented with 13 series of wet days (Sect. 2.2). Note that these categorical rain records do not go as far back as 1728, but mostly only back to 1864 (see Table 1). In order to better illustrate the effect of adding categorical data, we decided to use all available wet day series for both experiments.

For the 1728 station set without wet day series, the overall accuracy is estimated at 77.8 % (see also Table 3). By adding wet days, this increased by 0.5 % to 78.3 %. While for the autumn and winter months, the accuracy increased by 1 %, it declined by 0.5 % for the summer months. For the 1864 station set, adding wet days to the predictors decreased total accuracy by 0.8 % to 86.5 % (compared to 87.3 % without wet days). Also, all seasonal accuracies show a decrease between 0.4 % and 1.3 %. This shows that adding wet day series to the model input leads to negligible changes in accuracy which are mostly within the range of uncertainty of model training. With very few pressure and temperature records available (i.e. for the 1728 station set), wet days can provide supplementary information for WT classification. However, in our case improvements were limited to autumn and winter where precipitation is largely determined by large–scale circulation, whereas for summer, the results are slightly less accurate when including rain observations, which is arguably linked to precipitation being more frequently driven by local convection. If abundant pressure and temperature series are available (i.e. for the 1864 station set), using wet days as predictors yields no benefits. In this context, we decided to omit wet day series for our final CAP9 reconstructions in Sect. 3.3.

## 3.3 Reconstructing CAP9 weather types 1728–2020

### 3.3.1 Model Performance and Reconstruction Quality

With the feedforward Neural Network (NN) outperforming the other approaches (Sect. 3.1), we extended the current WT series for the CAP9 classification back to 1728. In order to provide an estimate for the model performance and by that of the reliability of our CAP9 reconstructions, a validation procedure as described in Sect. 2.4 was applied. The station series that have been used as predictors are described in Sect. 2.2. A summary on the resulting model architectures can be found in the supplement (Sect. S.4). Table 3 gives an overview of the validation results in the form of overall accuracy and average HSS



for predicted CAP9 WTs vs. the original predictand time series (1957–2020) by MeteoSwiss for all station sets. Results are again given for the whole period and distinguished by season. The accuracy for the earliest period between 01.01.1728 and 31.12.1737 is already remarkably high with a value of 77.8 % despite the limited set of available stations. Adding more station series generally improves the accuracy and skill score values (with some remaining variability depending on model training

runs). Whereas reconstructions for most station sets show slightly less skill and lower accuracies for the summer months (JJA), differences to the overall average remain small with values of approximately 1 % for accuracy and 0.1 for the HSS.

Table 3: validation results for the feedforward NN models with different station sets (named after their start year). Acc = average accuracy, HSS = Heidke skill score. Shown are estimates over the whole year (ANN), and the individual seasons (winter: DJF, spring: MAM, summer: JJA, autumn: SON)

| Station Set | ANN | DJF | MAM | JJA | SON |
|---|---|---|---|---|---|
| 1728 | Acc = 77.8 | Acc = 78.9 | Acc = 77.0 | Acc = 77.8 | Acc = 77.6 |
| | HSS = 0.76 | HSS = 0.75 | HSS = 0.75 | HSS = 0.69 | HSS = 0.74 |
| 1738 | Acc = 78.9 | Acc = 80.0 | Acc = 78.2 | Acc = 79.5 | Acc = 77.8 |
| | HSS = 0.77 | HSS = 0.77 | HSS = 0.77 | HSS = 0.72 | HSS = 0.75 |
| 1749 | Acc = 82.8 | Acc = 84.0 | Acc = 82.7 | Acc = 81.6 | Acc = 82.9 |
| | HSS = 0.81 | HSS = 0.81 | HSS = 0.81 | HSS = 0.73 | HSS = 0.80 |
| 1756 | Acc = 83.2 | Acc = 84.3 | Acc = 82.8 | Acc = 82.8 | Acc = 82.9 |
| | HSS = 0.82 | HSS = 0.82 | HSS = 0.82 | HSS = 0.78 | HSS = 0.80 |
| 1764 | Acc = 84.8 | Acc = 85.6 | Acc = 85.2 | Acc = 83.4 | Acc = 85.1 |
| | HSS = 0.84 | HSS = 0.83 | HSS = 0.84 | HSS = 0.76 | HSS = 0.83 |
| 1771 | Acc = 83.9 | Acc = 83.8 | Acc = 83.9 | Acc = 83.6 | Acc = 84.4 |
| | HSS = 0.83 | HSS = 0.81 | HSS = 0.83 | HSS = 0.75 | HSS = 0.83 |
| 1781 | Acc = 84.8 | Acc = 84.6 | Acc = 85.0 | Acc = 84.8 | Acc = 84.8 |
| | HSS = 0.83 | HSS = 0.82 | HSS = 0.84 | HSS = 0.77 | HSS = 0.83 |
| 1790 | Acc = 84.7 | Acc = 84.8 | Acc = 84.8 | Acc = 84.3 | Acc = 84.9 |
| | HSS = 0.84 | HSS = 0.82 | HSS = 0.84 | HSS = 0.77 | HSS = 0.83 |
| 1818 | Acc = 84.3 | Acc = 84.1 | Acc = 84.6 | Acc = 83.9 | Acc = 84.7 |
| | HSS = 0.83 | HSS = 0.81 | HSS = 0.83 | HSS = 0.73 | HSS = 0.83 |
| 1864 | Acc = 87.3 | Acc = 87.6 | Acc = 87.8 | Acc = 86.9 | Acc = 87.0 |
| | HSS = 0.86 | HSS = 0.85 | HSS = 0.87 | HSS = 0.82 | HSS = 0.85 |

To provide more insight into the patterns of correctly and wrongly classified WTs and the reasons why the model is not able to assign certain WTs correctly, further analyses have been performed. Fig. 3 shows the confusion matrices for the station sets

1728 and 1864 for the reference period. Whereas accuracies may vary between the models, training runs and station sets, the actual WTs that are wrongly assigned for each true class are similar. For a true WT 8 most false predictions show WT 5, and for WT 9 most false predictions show WT 7. Already Schwander et al. (2017) found these two pairs hard to distinguish, leading them to reduce the number of WTs accordingly. However, this confusion does not necessarily hold reciprocally, as WTs 5 and



7 represent a weaker form of WTs 8 and 9, respectively, and are more likely to be confused with other similar WTs (see Fig. 380 3).

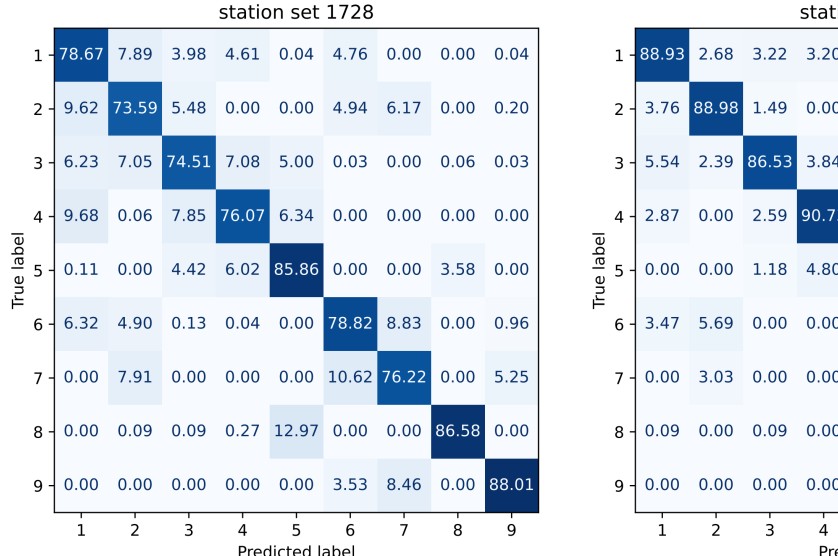

**Figure 3.** confusion matrices for reconstructions (columns) with station sets 1728 (left) and 1864 (right) against reference CAP9 series (rows) for the reference period. Values are given in percent of the respective WT occurrence.

Figure 4 shows the patterns of pressure deviations from the average of the time series (in standard deviations) for each station and weather type within the reference period. Indicated are the average values for correctly assigned (blue) and wrongly assigned (red) WTs, as well as the range between the 5 % and 95 % quantiles (shaded areas) from the reconstruction with the 1864 station set. Evidently, some WTs have very similar patterns with a large overlap (e.g. WT 5 and WT 8) making a
distinction difficult. For most WTs dominated by extremely high or low pressure (e.g. WTs 5, 8, and 9), wrongly assigned WTs are linked to more moderate values in the pressure data. Furthermore, regional differences in the pressure distribution can be identified as a source of error. For example, WT 6 is more likely to be confused with other WTs for days with stronger low pressure systems over northern Central Europe. Such regional patterns can also be found for WTs 3, 4, and 7. The corresponding temperature profiles (see supplement Fig. S5.1) show similar patterns with observed temperatures for days with wrongly
assigned WTs closer to the mean (WTs 2, 3, and 6) or regional differences (WTs 7, 8, and 9), although these patterns are much less distinct. The same evaluation for the other station sets provides similar results (not shown).

Figure 5 shows average sea level pressure maps for the period 1957–2020 derived from ERA5 (Hersbach et al., 2020; Bell et al., 2021). The maps are separated by season, namely winter (DJF, Fig. 5a) and summer (JJA, Fig. 5b), as well as by reference
series (top), correctly attributed WTs (centre) and false predictions (bottom). Note that WT 8 does not occur during the summer



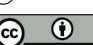

**Figure 4.** station data pressure patterns for correct (blue) and false (red) predictions from the 1864 station set for all nine WTs. Shown are average (lines) and the 5 % – 95 % quantile interval (shaded areas) in units of standard deviations.

months (see the seasonality in Fig. 1, as well as Fig. S5.2 in the supplement) and that no day was wrongly assigned to WT 9 in the reference period, hence the empty panels in Fig. 5b. Whereas false predictions for the winter months are strongly dominated





by weaker–than–average pressure distribution rather than regional shifts, results are less clear for the summer months. Whereas slight regional shifts are apparent (e.g. for WTs 1, 3, and 7), the reason for false predictions in summer seems to originate from

other sources, arguably patterns in temperature or general difficulties of the model to capture the smaller pressure gradients in this season.

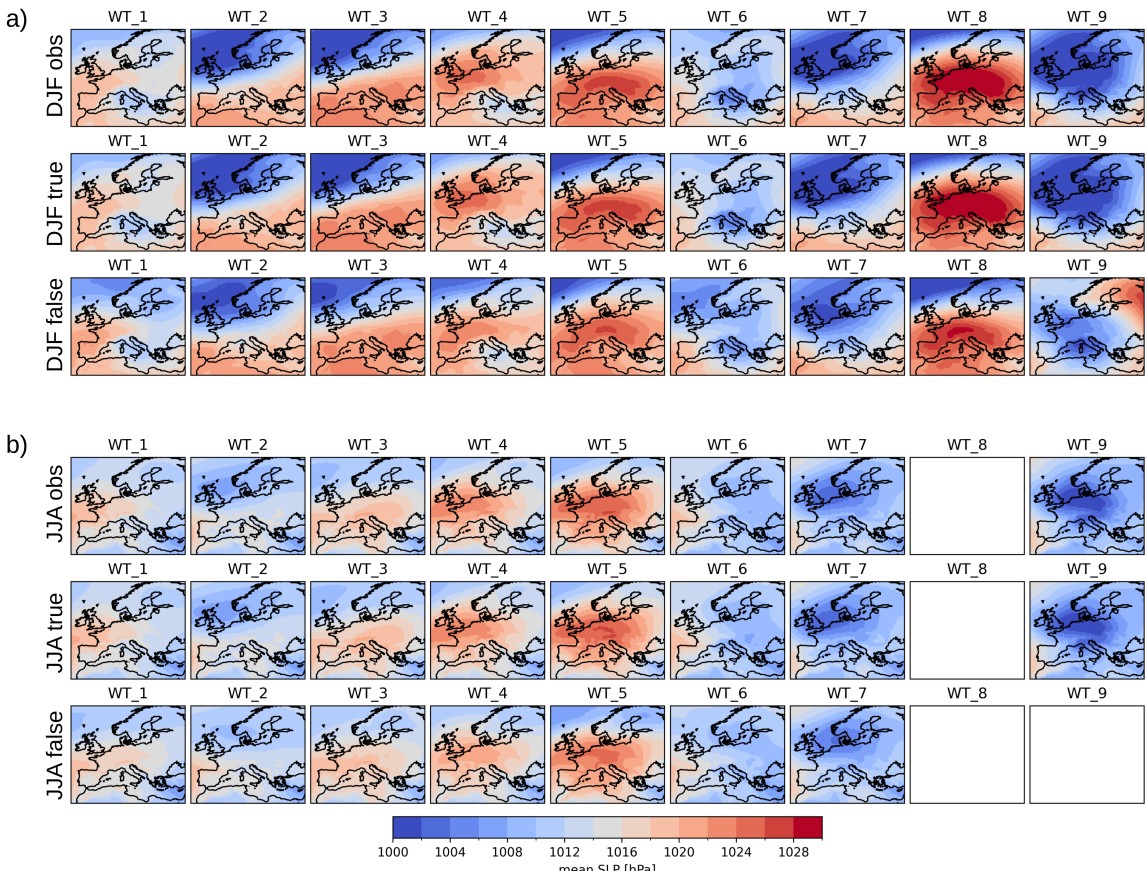

**Figure 5.** climatological average of sea level pressure 1957–2020 for CAP9 WTs for a) the winter and b) the summer months. Shown are the averages according to the official WT series by MeteoSwiss (top, obs), correctly predicted WTs (centre, true) and wrongly predicted WTs (bottom, false)

As the synoptic circulation is constantly changing, weather types might change over the course of one day. This has to be taken into account when analysing daily weather type reconstructions, as such WT transitions may be a source of error. In the reference CAP9 series, 19.1 % of days are persistent weather situations with the same WT on the days before and after.

A majority of days (46.4 %) is a partly transient situation with the same WT on one of the neighboring days and a different one on the other and in 34.5 % of the cases, different WTs occur on both neighboring days (transient situation). Taking





reconstructions using station set 1864 as an example, the correctly classified WTs show the same percentages. For the days with false predictions, however, transient WTs are overrepresented (48.0 %), whereas only 7.6 % show persistent conditions. We can conclude that transient WTs play an important role as a source of uncertainty in daily WT reconstructions. The chosen WT for these cases might be arbitrary depending on slightly stronger patterns (i.e. dominating by a small margin) visible in the daily averages of station observations. This issue might be solved by introducing a neutral (transient) class or by calculating WTs for a specific time of the day (e.g. 12:00 UTC) using subdaily data which is, however, less readily available for the early instrumental period.

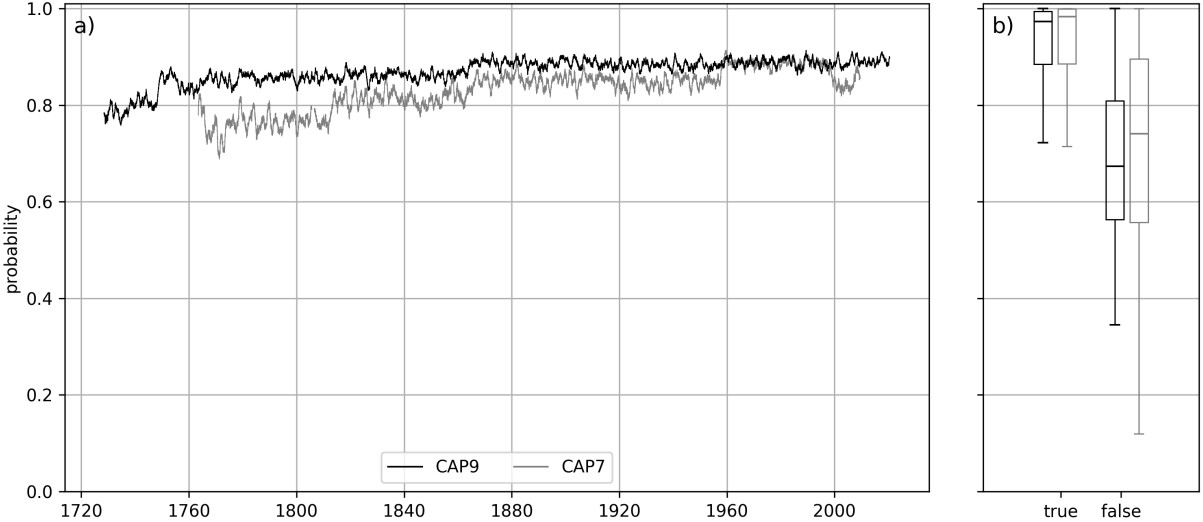

**Figure 6.** a) 365–day running mean of the daily maximum probability (fraction) of the reconstructed CAP9 WT series (in black) and CAP7 WT series by Schwander et al. (2017) (in grey) and b) boxplots of the probability for correctly (true) and wrongly (false) attributed WTs within the reference period for the respective reconstructions.

A next interesting feature to look at is the confidence of the model in its predictions, i.e. the probabilities with which the weather types are classified. As stated in Sect. 2.3, for each day the NN attributes a probability to all WT classes and the respective class with the highest probability is selected as the predicted (or most likely) WT. Figure 6a shows a one–year running mean of the daily probabilities of the predicted WTs (in black) for the whole period of reconstruction. It shows values around 0.8 in the first two decades, increasing to values between 0.825 and 0.875 in the middle of the 18[th] century and to values between 0.85 and 0.9 in 1864. Compared to the CAP7 reconstruction (in grey; see also Schwander et al., 2017, their figure 4), the NN classification shows higher probabilities and less variance pointing to a high consistency of the reconstructions over time. The distinction of daily maximum probabilities by correct and false classifications in the reference period (Fig. 6b) reveals that the model used for our CAP9 reconstruction is less confident for WTs that were wrongly assigned (median = 67.4 %) than for correct attributions (median = 97.3 %). This is in line with the above finding on transient WTs that mixed signals in




the surface observations may lead to false classifications. A comparison with the probabilities from the CAP7 reconstructions
shows the same pattern, although the SMD approach is more confident for false classifications on average.

### 3.3.2 The new CAP9 Reconstructions in a Climatological Context

In this section, we look at the CAP9 WT reconstructions produced with the chosen NN approach (Sect. 2.3) for the full period
1728–2022. The aim is to analyse their quality and consistency, i.e. look for possible discontinuities in WT frequencies, as they
have e.g. been found for the Hess & Brezowski WT classification in the mid–1980s (Mittermeier et al., 2022). Furthermore,
we compare occurrence frequencies of reconstructed CAP9 WTs with the CAP9 reference series and CAP7 reconstructions on
climatological timescales to analyse the representation of internal climate variability of WTs in the past decades to centuries.

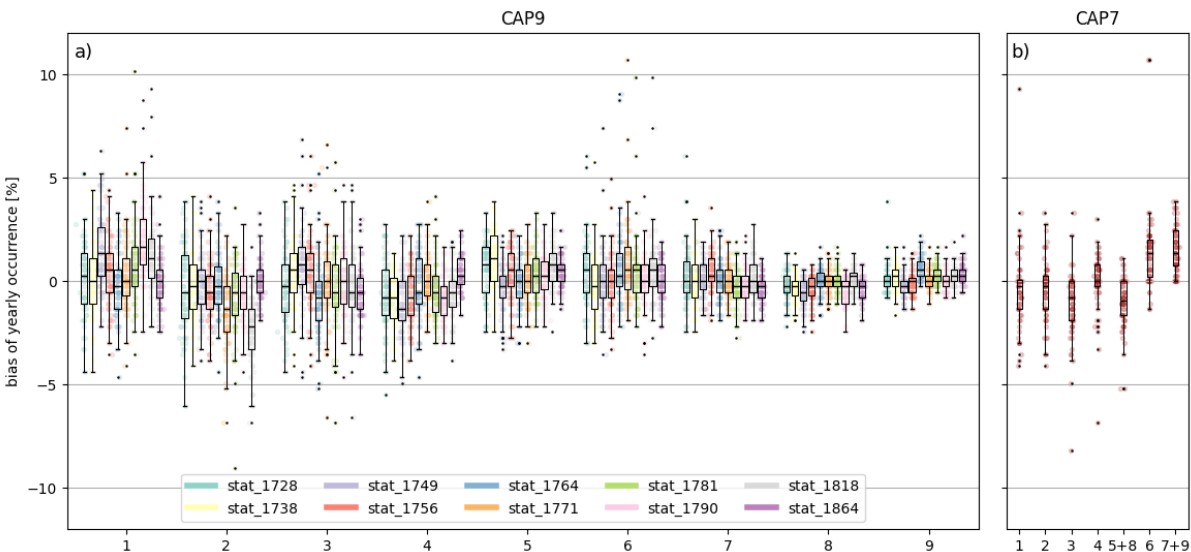

**Figure 7.** Bias of yearly WT occurrence (in % of days) for all WTs (x–axis) and station sets (colors) in a) the NN reconstruction and b) the
CAP7 dataset by Schwander et al. (2017).

An important quality characteristic are biases in the occurrence of different WTs. Figure 7 illustrates the percentual bias in
yearly WT occurrence for the reference period (n = 63 years) separated by station set and weather type. For comparison, the
biases were also calculated for the original CAP7 reconstructions by Schwander et al. (2017) (Fig. 7b), although these values
might underestimate the actual bias as discussed in Sect. 2.4. The median biases remain within 1–2 % for all weather types
and station sets with no systematic over– or underestimation of an individual weather type. Some outlier years are evident for
WTs 1, 3 and 6 (overestimation), as well as WTs 2 and 3 (underestimation). Compared against CAP7 WTs (representing the
most dense of the available station networks), a larger spread of values can be observed for some of the early station sets of
our reconstructions. A more equal comparison with station set 1864 shows reduced biases compared to CAP7 for most WTs



and less pronounced outliers. The feed–forward NN improves the bias values especially for the rare WTs (7, 8, 9, as well the slightly more frequent WTs 5 and 6; see Fig. 1).

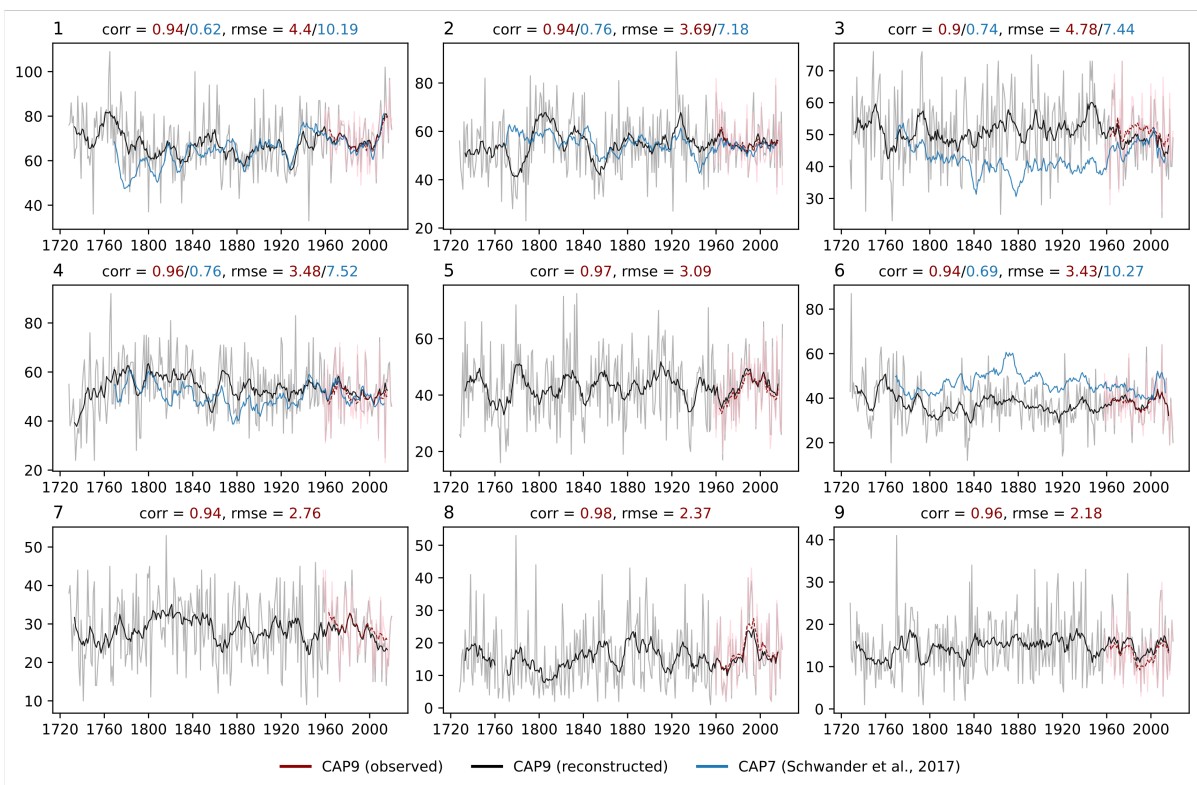

**Figure 8.** Yearly occurrence of reconstructed CAP9 WTs (lighter colors) with 10–year running mean (darker colors). Shown are the CAP9 reference series (red), the CAP9 reconstructions (black), and the CAP7 reconstructions (blue). Indicated are correlation and root mean squared error for the yearly WT occurrence with respect to the reference series.

Figure 8 illustrates the full reconstructed time series of the yearly WT occurrence for each weather type (in black), again with the CAP9 reference series (in red) for comparison. For better readability, a 10–year running average (including the CAP7 reconstructions in blue) is indicated. In general, our CAP9 reconstructions correspond better to the reference series as the pre-

vious CAP7 reconstructions. The yearly WT occurrence in our new CAP9 reconstruction shows higher correlations (average = 0.948) and lower root mean squared error values (average = 3.35 days) than the CAP7 reconstructions (corr = 0.77, rmse = 8.26 days) by Schwander et al. (2017). A positive bias for WTs 6 and 9, as well as a negative bias for WT 8 determined in Fig. 7 can also be seen in the time series. Further back in the past, CAP7 and CAP9 reconstructions have some discrepancies, especially for WTs 3 and 6. Considering the biases found for CAP7 (Fig. 7), these differences can be attributed to an un-

der(over)–estimation of WT 3(6) in these reconstructions. The results presented in Fig. 8 suggest that the reconstructed CAP9 time series do not show any apparent artificial discontinuities that go beyond natural variability.







**Figure 9.** 10–year running average of yearly weather type occurrence by season and weather type. Shown are the CAP9 reference series (dotted lines), the CAP9 reconstructions (solid lines), and the CAP7 reconstructions (dash–dotted lines).

More detail on the occurrence frequency is given in Fig. 9, where we show the 10–year running average yearly WT occurrence distinguished by season. These seasonal occurrence patterns of CAP9 reconstructions generally match the occurrence in the reference series. For WTs 6 and 9, the observed positive bias in the reconstructions can be mainly attributed to an overestimation of WT occurrence in spring (MAM). The negative bias of WT 8 on the other hand is linked to an underestimation of this WT in the winter months (DJF). Seasonal patterns of the CAP7 reconstruction show less consistency with the reference series and even some arguably artificial trends (e.g. WT 1 in summer) can be detected. The attribution of a bias to seasonal differences points to an important issue in WT reconstruction. As most WTs (1, 5, 7, 8, and 9) show a pronounced seasonality they can be difficult to predict for a model that is trained over all seasons. Tests with training individual models for each season improved the results, although for some WTs (e.g. WT 8 in summer), the available sample for model training becomes too



small. Another option might be to include seasons or months as categorical predictor variables, although this has not been tested in this study.

## 4 Conclusions

In our study, we applied various supervised machine learning (ML) methods for station–based weather type (WT) reconstruction in order to assess their performance and to find an optimal ML approach for this purpose. With the model showing the best performance and using additional station observations, existing CAP9 WT series have been extended back to 1728.

Our results show that all ML approaches perform well when tested on the daily CAP7 WT classification. Independent estimates of accuracy and HSS show a better performance of all tested models compared to the common statistical classification approach used as a baseline. ML methods can thus indeed profit from their ability to detect non–linear patterns. The feedforward neural network slightly outperformed the other ML approaches and was therefore used to create the CAP9 WT reconstruction. The use of qualitative rain observations did not improve our reconstructions, but instead yielded partially worse results and was thus omitted for our reconstructions. The extension of the existing CAP9 classification back to 1728 constitutes a novelty in WT reconstruction. The resulting WT time series proves to be accurate in various facets. No artificial trends or discontinuities could be detected. The year–to–year variability and the seasonality of the WTs are well reproduced. Nevertheless, depending on the available set of stations, some over– and underestimation of WT occurrence could be determined. Our results emphasise the importance of constantly improving WT classification methods with new options and data available.

Some challenges or limitations of our approach persist. First, the station data availability is usually scarce in the early instrumental period. Further data rescue efforts may provide additional observations at important locations for WT classifications. Although our experiment with adding qualitative rain information did not improve the reconstructions, other qualitative information more directly linked to large–scale circulation such as wind direction might lead to improvements. Unfortunately, the availability of digitised, long–term wind direction records is sparse and therefore could not be assessed in this study. A second challenge is the number of samples of each WT in the reference series. WTs with low occurrence frequencies and strong seasonality can pose a challenge for our WT reconstruction approach. Adding seasons as additional predictors or training different models per season could solve this issue, although the sample size of rare WTs might be too small. Also in general, the size of the training dataset has to be proportionate to the number of WT classes in order to find robust model weights and biases. A third issue is the daily resolution of input and WT data: transient weather types leave a mixed signal in the daily average observations making the distinction on a daily resolution difficult. This issue might be solved with the use of subdaily data which are, however, less readily available in the form of long and homogeneous time series.

Our CAP9 reconstruction represents the longest daily WT series available and allows for studying decadal circulation variability in the context of past climatic changes, as well as the impacts of associated synoptic situations on the surface, e.g.





extreme events. On the methodical side, future research may focus on including wind direction observations to improve and
extend WT reconstructions even further back in time, although this requires tremendous digitisation efforts. Whereas we fo-
cused on reconstructing CAP9 WTs, our ML models may be adopted to other WT classifications and regions.

*Code and data availability.* Most station series used are publicly available on data repositories (Brugnara, 2022; ECA&D, 2024; GeoSphere
Austria, 2021; DWD Climate Data Center, 2024). Observational records and weather types provided by MeteoSwiss can be directly obtained
from MeteoSwiss on request. The reconstructed CAP9 WT series, as well as the corresponding code for model building and training is
publicly available at the BORIS repository (Pfister, 2024, https://doi.org/10.48350/195666)

*Author contributions.* LP had the idea and planned the campaign with contributions from LW and SB. YB and NI provided observational
data and code for homogenisation; LP and LW performed the computations, provided the visualisations, and wrote the manuscript. LW, SB,
YB, and NI reviewed the manuscript.

*Competing interests.* The authors declare that they have no conflict of interest.

*Acknowledgements.* The authors would like to thank all the institutions that provided the valuable meteorological station observations (Me-
teoSwiss, DWD, GeoSphere Austria, KNMI). Particular thanks goes to Mikhaël Schwander for providing the original input data he used for
his CAP7 reconstruction, as well as to Luis Rivero who performed insightful preliminary research testing neural networks for weather type
reconstruction.

*Funding.* Lucas Pfister and Noemi Imfeld were funded by the Swiss National Foundation SNSF project "Daily Weather Reconstructions to
Study Decadal Climate Swings". Additional funding for Yuri Brugnara and Lucas Pfister was available through the "Swiss Early Instrumental
Meteorological Data" (CHIMES) project funded by SNSF and the "Long Swiss Meteorological Series" project funded by the Global Climate
Observing System (GCOS) Switzerland. Lena Wilhelm was funded by the Swiss National Science Foundation (SNF) Grant CRSII5_201792.



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
