# Peer review of "Weather Type Reconstruction using Machine Learning Approaches"

_EGUsphere, 2024_

## Referee Comment (RC3)

**Review of „Weather Type Reconstruction using Machine Learning Approaches" by Pfister et al.**

**Summary**

The authors assess various machine learning approaches for weather type reconstruction and then use the best-performing model to extend the time series of weather types over central Europe back to 1728. I consider the study a worthwhile addition to the effort of reconstructing past weather, but the authors should first address some questions that I raise below.

**Major comments**

The authors try to make their results comparable with the study by Schwander et al. (2017), which has a marked negative impact on the readability of the text (e.g., in 81, 108, 150, 247, 268, etc.). Repeated efforts to explain differences in the dataset and methods, especially the constant switching between the CAP7 and CAP9 methods, lead to unnecessary confusion for the reader. Additionally, if I understand correctly, different classifications were used in validation and reconstruction, which could be considered questionable, to put it mildly. I suggest moving the discussion/comparison of the authors' results with the other study to one paragraph in Section 3, including Schwander's method among the methods they validate, and consistently using either the CAP9 or the CAP7 method. Schwander's model was already recalculated by the authors (and an error in the original calculation was found), which means that the change will be relatively easy.

In Conclusions, the authors should mention that best performing method differs depending on season, used data, and validation metric, instead of the over-generalizing "The feedforward neural network slightly outperformed the other ML approaches…" and that other method(s) lead to comparable results in a shorter time. I would also consider worth stating here that the ML approaches may be less sensitive to the quality of the classification (see my suggestion bellow, l. 306).

**Minor comments**

„in the accuracy of the used methods" is rather vague

CAP9 abbreviation is used but not defined

WT abbreviation is defined here but not used consistently in the paper

Can you refer to a global-scale evaluation of the skill of classifications that would confirm your claim that „In Europe classifications prove particularly useful to describe the prevailing atmospheric conditions?"

Given your references here, probably "model outputs" would be more appropriate and clearer than "weather forecasting model simulations"

"introduced, that" –> "introduced that"

"With the newest generation of reanalysis datasets, many WT records could already be extended back to the 19th century (Philipp et al., 2010; Jones et al., 2014)." How do these two rather old references support your claim regarding the newest reanalyses?

Maybe "machine learning" would be better than „artificial intelligence"?

50-55 First, you write that "artificial intelligence is commonly used for classification…in climatological research" and then "In the context of WT classifications, ML is still a rather novel approach." Please clarify. Furthermore, cluster analysis is commonly accepted as a machine learning approach and some clustering methods are among the oldest WT classification methods. Therefore, I would oppose your latter claim.

It is not clear what "this pioneering work" refers to

Please add a little bit more detail to the decomposition; for example, what similarity matrix is decomposed?

I'd suggest "the first step… the second step" rather than "a first…a second"

Fig. 1 Some stations in the figure (e.g., Cadiz) are not visible. I suggest moving the symbols in front of the coastal lines and increasing the size a little

ERA5 goes back to 1940, why was only 1957–2020 used?

Instead of "are well distributed across most parts of Europe" I would suggest a considerably more accurate "are relatively well distributed across central Europe", or similar.

Probably "SLP" instead of "pressure" would be clearer and more accurate. This occurs several times throughout the paper

It is not clear what "the latter study" refers to

Is not "The station sets used for the reconstruction of CAP9 WTs (Sect. 3.3.1) are summarised in Fig. 2" at odds with "Figure 2. Station sets of a) pressure and b) temperature used for the model comparison", considering that CAP7 was used for comparison?

"Thus, temperature data were corrected for their seasonality by fitting the first two harmonics to each temperature record, which was then subtracted from the data." should be made clearer; your sentence suggests that the temperature record was removed. I suggest "Thus, temperature data were corrected for seasonality by fitting the first two harmonics to each temperature record and then subtracting these harmonics from the data."

"their contribution" is not clear

"they are trained" is not clear

"trained on the same station data" is surprising to me, since above you explained considerable differences between your dataset and that used by Schwander et al.

"both, hyperparameter" –> "both hyperparameter"

Table 2 Is it necessary to repeat "Acc =" and "HSS=" for all values? Did you consider another type of graphical output, which may be more readable?

"This best–performing model" is not clear

306-309 A useful interpretation could be that ML models may be less sensitive to the used classification compared with the simple baseline model. One may argue that projecting summer circulation on 7 WTs trained for an annual time series is far from ideal, because the classification lacks the necessary detail to explain the relatively weak and specific summer circulation. One reason for this is the dominant WT1: if a stand-alone classification was trained by cluster analysis only for summer days, it would find relevant summer patterns that would be more or less equally populated instead of the snowballed WT1.

"the overall atmospheric signal seen in a combination of information" is rather vague and unclear

I would also consider adding "western Europe" because (south)westerly advection is an important feature of circulation over central Europe

I think that "The model comparison revealed the feedforward neural network (NN) to exhibit the highest accuracy and HSS estimates" is an exaggeration and at least "on average" should be added into the main clause. However, to me it seems that the NN, RNN and CNN lead to nearly identical results and Table 2 shows that the best performing method is sensitive to the choice of season, data and validation metric.

377-9 "For a true WT 8 most false predictions show WT 5, and for WT 9 most false predictions show WT 7. Already Schwander et al. (2017) found these two pairs hard to distinguish, leading them to reduce the number of WTs accordingly." I do not think that the results shown in Table 3 support Schwander's claim. First, accuracy for WTs 8 and 9 are not worse than that for the other WTs. Second, these two WTs are outliers – if you imagine the 9 WTs ordered such that their position (for instance in a 1D or 2D Sammon map, the first PC plain, etc.) respects their position in the high-dimensional space, the two outlying WTs would simply neighbour with fewer WTs. Consequently, most (all) of the false hits will be linked to the one "closest" WT (which has a strongly correlated but weaker circulation pattern). I would not even consider this an artefact of the methodology but rather a geometric necessity.

Figure 3: Consider specifying what are the "reference period" and "reference CAP9 series" in this case. In the validation phase, it is clear what accuracy means. However, this is not the case for the cross-validation periods

Paragraph starting at 380+Figure 4 This could use some additional introduction and explanation – you lost me here

Figure 5 Showing true and false maps as deviations from obs would probably support your interpretation more clearly

408+409+488 Consider "transient situations" or similar instead of "transient WTs". Transient WTs are WTs with low persistence, but you did not show that.

"The chosen WT for these cases might be arbitrary depending on slightly stronger patterns (i.e. dominating by a small margin)" Please reword, I do not follow

"or by calculating WTs for a specific time of the day" I do not believe that this would have any effect on the presence of transient/boundary circulation fields in your data. Circulation fields form a continuum of patterns and I do not know of any reason to expect that instantaneous and averaged fields differ in this respect. Ditto 489

Figure 6b boxplots add y-axis labels (%?) and explanation of shown percentiles

Figure 8 and especially 9: Consider using an even longer filter, the lines are still very noisy

"The results presented in Fig. 8 suggest that the reconstructed CAP9 time series do not show any apparent artificial discontinuities that go beyond natural variability." How was this tested? You present this in Conclusions as one the main results but you did not provide any testing

"Our results emphasise the importance of constantly improving WT classification methods…" One may argue that you did not test or enhance the CAP9 methodology itself, therefore rewording the sentence may be advised

"occurrences" or similar may be better than "samples"

"Adding seasons as additional predictors or training different models per season could solve this issue, although the sample size of rare WTs might be too small." If season-specific classifications are trained (and a suitable classification method utilized that does not tend to identify marginal WTs), the issue of WTs with marginal occurrences will disappear. Additionally, season-specific classification could have fewer WTs, which would (I suppose) decrease the computation cost

The availability of monthly gridded datasets is mentioned in the paper. I was wondering whether including monthly SLP patterns as one the predictors could improve the models.

I would also welcome a note on the possible sensitivity of the models to the chosen classification and its parameters. Classification method is one of the major factors of any synoptic-climatological study and I would expect a significant sensitivity of WT reconstructions (and validation metrics) to changes in the classification methodology.

---

## Author Comment (AC1)

**Review of „Weather Type Reconstruction using Machine Learning Approaches"**

**General comments:**

This study uses machine learning methods to reconstruct the CAP9 weather type classification for Europe back to the year 1728 based on station observations. Four different machine learning methods are tested (multinomial logistic regression, random forest, feedforward neural network, RNN/CNN) and compared to a reconstruction method based on Mahalanobis distance and to the original CAP9 time series published by MeteoSwiss for the reference period 1957-2020.

I find this study to be interesting, well structured and well written, and with high scientific quality of the methods and results presented.

My main objective is that the scientific relevance should be better emphasized. The authors should better explain why a weather type classification based on station observations is beneficial, especially in light of the gridded EKF400v2 reanalysis product, which goes back to the year 1602.

We'd like to thank the reviewer for this detailed review and generally positive assessment of our manuscript.

Regarding the scientific relevance, long-term WT reconstructions like the one presented in our work allows to trace decadal to multi-decadal variability and long-term changes of synoptic circulation patterns over 300 years (indicated in L. 30 f). Furthermore, with the link between WTs and surface processes, our work opens the door for a great amount of climatological analyses, e.g. for deriving a flood probability index (see Brönnimann et al., 2019) or other weather extremes.

As described in lines 32 ff in the manuscript, reanalysis datasets may be very well used for WT reconstruction. The issue, however, is the resolution of the input data which must be daily or even finer in order to allow reconstructing a classification of synoptic atmospheric patterns. EKF400v2, as well as many other reconstructions of past weather and climate going beyond the 19th century, only provide monthly information. For WT reconstructions going this far back in time, station observations or weather diaries are the only suitable sources of information available.

We'll try to emphasize this better in the revised manuscript, together with the scientific relevance of our work in general.

**Specific questions**

Line 47: "Whereas common statistical approaches seem to have reached their limit for this purpose, (…)". Why have they reached their limit? Please explain this better.

Thank you for this question. The limitations refer to previous WT reconstructions using common statistical approaches described in L. 38 ff. We'll rephrase this sentence in the revised manuscript: „Whereas common statistical approaches have been effective in capturing prominent atmospheric patterns, their ability to handle more complex, non-linear relationships and incorporate qualitative data is limited. Supervised machine learning (ML) classification methods offer a promising alternative, as they are well-suited for recognizing intricate non-linear patterns in atmospheric variables."

Figure 1 (Right): What is the unit of the average monthly occurrence? Days or counts?

Thank you. We'll indicate the units (number of days per month) in the y-axis label of Fig. 1 in the revised manuscript.

Table 1: Please explain what exactly the temporal pressure gradient is and how it is derived from the historical station observations.

The temporal pressure gradient is explained in L. 126 ff. We'll add a reference to Table 1 and the variable $\triangle$p at these lines to make this clearer in the revised manuscript

Line 32ff./Line 154: I was a bit surprised to learn about the EKF400v2 reanalysis product in Line 154, which covers the period 1603-2003 and was not mentioned during the Introduction. What is the point of deriving weather type classification from station observations if a gridded reanalysis product is available for the earliest period of your observations and even before? This is in direct contradiction to the statements in line 32ff. and thus to the motivation for this paper: "With the newest generation of reanalysis datasets, many WT records could already be extended back to the 19th century (…)." and "(…) the limit for WT classifications based on atmospheric fields is set by the 20th Century Reanalysis version 3 (…), which extends back to 1806". Please correct these statements in the Introduction and revise the motivation for a classification based on station observations in light of the available EKF400v2 reanalysis going back to 1603.

Thank you for this comment. As mentioned in our response to the reviewer's introductory comment, EKF400v2 unfortunately only provides monthly data and is thus not suitable for reconstructing daily weather types.

Line 193: Which variables are the five predictors?

Thanks for this question. The limitation to five predictors refers to a general limitation of the number of predictors to avoid multicollinearity and overfitting, rather than an a-priori choice of certain variables. The optimal combination of predictor variables is only determined during model training and described in the results section (Sect. 3.1) in L. 293 ff.

Chapter 2.3.3 and 2.3.4: What is the structure of the input layer? The Appendix says 6,8,9 x None (Time). Are these the number of stations used? What variables are used? In general, I miss a better description of the input variables of the machine learning methods. Are temperature and pressure time series used at all stations? What about the temporal pressure gradient? Please specify the structure of your input layers.

Thank you for this comment and thank you also for having a close look at the supplement. The structure of the input layer of the feedforward neural networks can be understood as a table of time series with one dimension (columns) with a length equal to the number of time series used as input (all stations and variables, e.g. 6 for the 1728 station set, see Fig. 2) and the other one with a length equal to the length of the time series (or batch). Your comment brought to light an important point that unfortunately went missing during the iterations of reworking our manuscript: whereas for the model intercomparison in Sect. 3.1 we used temporal pressure gradients as input (see L. 126 f) consistent with the baseline approach, those gradients were omitted for the NN used for WT reconstruction (Sect. 3.2 and 3.3) as tests (not shown) did not reveal consistent improvements by adding this variable. We will reinsert this statement in Sect. 2.2 and describe the input variables of the individual ML methods in Sect. 2.3 in the revised manuscript.

Chapter 2.3.3 and 2.3.4: Is the lat/lon information of the stations used as input as well? Does the machine learning model have any information on the position of the time series? If not, please discuss this.

Thank you for this question. We did not include lat/lon information or other direct information on the station location as input to the model. As supervised machine learning methods are designed to identify patterns in the input data (i.e. station observations) related to a given category (in this case with given circulation patterns (WTs) over a certain region), we expect the models to find the relevant spatial patterns even without the knowledge of the exact position of the stations. Tests were made with indirect spatial information, using spatial gradients between stations (e.g. the pressure difference between Stockholm and Milan for a north-south gradient). However, they did not show any benefits with the tested models (not shown) supporting the former statement. From the presented methods, CNNs would be the most appropriate to include the spatial dimension directly, treating the station observations as cells of a spatial grid. However, this would need further research related e.g. to the grid structure and imputation of missing cells. As the validation metrics showed good results without spatial information, we did not pursue this issue in our study.

Line 241f. What are "(…) all available pressure and temperature series"? Please specify.

Thanks for this question. As stated in L. 240 f, we tested subsets of stations or variables (e.g. only pressure / only temperature, subset of stations to achieve a more equal spatial distribution). „All" in this case means that we use the full set of stations and variables available. We will try to make this clearer in the revised manuscript.

Line 278: What is the advantage of the Heidke skill score? How can it be interpreted compared to overall accuracy?

The Heidke skill score includes an important aspect that the overall accuracy alone does not indicate: as described in L. 280 ff, the HSS is calculated for each WT individually and thus accounts for differences in the occurrence frequency of the individual WTs. Overall accuracy, however, may weight more frequent WTs stronger. Therefore, as an example, a high accuracy together with lower HSS values allows for the interpretation that prediction errors might originate from individual WTs. Furthermore, the HSS provides a value with respect to a reference (forecast by chance).

Line 353: "(…) which are mostly within the range of uncertainty of model training." How do you quantify the range of uncertainty of model training to reach this conclusion?

Thanks for this remark. As a rough measure for the uncertainty of model training we took the variance of the validation metrics of the dataset splits (outer folds). The variance of accuracy and HSS in the outer folds were larger than the improvements gained by adding wet days as additional predictors. We'll indicate this in the revised manuscript.

Line 368f.: "The accuracy for the earliest period between 01.01.1728 and 31.12.1737 is already remarkably high with a value of 77.8 % despite the limited set of available stations." This sentence is misleading, because it suggests that you know the accuracy of your model for the earliest period. But you can't estimate the accuracy of the early period, because you don't have labels for that time to which you could compare your classifications to. If I got it right, the 77.8% indicate the accuracy of your trained model for a test set from the period 1957-2020 compared to the MeteoSwiss time series, whereby your model uses the number of stations only that are available since 1728. But your actual accuracy in the early period could be lower than that due to lower data quality in the early period e.g. measurement errors. Please refine the statement and discuss the data quality within your time series.

This is an excellent comment. We agree that the phrasing of this sentence may be misleading. We will change this in the revised manuscript and add a sentence on the accuracy in the reference period vs. the actual accuracy in the past at L. 370: „The achieved accuracy using the smallest station set (stations available from 01.01.1728 to 31.12.1737) is already remarkably high […]. Adding more stations […]. Note that validation metrics shown in Table 3 only provide values with respect to the reference period 1957-2020. The actual values for the past periods may be lower due to larger uncertainties and errors in the data, but unfortunately cannot be determined due to the lack of a historical reference WT series."

Figure 5: The plots are quite small and hard to compare by eye. It could help to increase the size and/or to show the differences of the false composites and the true composites to obs composites in order to better show the differences in the pressure fields. I'm also wondering how many cases each composite plot is derived from. The numbers could be indicated above the plots.

Thank you for these suggestions. We will increase the size of the plot and indicate the number of cases in a revised figure. We originally also considered showing deviation maps of true / false composites with respect to observations (see also response to RC3). However, we found this to blur information on the position of low and high pressure systems in the true / false prediction maps. Although less apparent than when showing deviations, the discussed differences between the maps can still be determined from the absolute values shown in Fig. 5, thus we deemed this solution to be better.

Discussion: I miss a discussion on why including the previous days in the RNN/CNN setup didn't help to improve the accuracy of the weather type classification. Is this in line with what the authors expected? What could be the reasons for this?

Thanks for this remark. Whereas we expected some improvement when taking the previous days into consideration, the available data for a certain day including the pressure gradient with respect to the day before (as used in NN) seems to be sufficient for correctly determining the corresponding WT. Added value of temporal series may be linked to information on preferential transitions which can complement station observations (arguably the case for few station series, see Table 2). We will add such a statement in L. 331 in the revised manuscript. The main sources of error (i.e. the reason for 10-20% wrongly assigned WTs), however, seem to have a different origin and cannot be solved by using data from previous days (e.g. the spatial coverage by stations, see L. 312 f, L. 480).

Supplement Table S2.1: Please explain the variable names

Thanks for this suggestion. We will explain the variable names in the table caption in the revised supplement.

---

## Author Comment (AC2)

**Review of "Weather Type Reconstruction using Machine Learning Approaches" by Pfister et al.**

**General comments**

The authors carry out a study for reconstructing Weather Types daily series back to the 1700s using a known WT classification with the use of several site measures and applying different Machine Learning Approaches to assess which one is more fit for the job.

The paper is well presented and written, however there are points that should be addressed:

1. The paper is based on an assumption that is not stated explicitly and cannot be taken for granted: Weather Types stemming from atmospheric variables can be assumed to remain the same across centuries. (See specific comment L.30-31). If we believe this hypothesis to hold, the authors made little effort to characterize temporal trends in the occurrence of the WTs and assess whether, from their reconstructed series, there have been shifts in occurrences from one season to another. From a climatic stand point I think these are relevant features of your very long (200+ years) classification.

   *Thank you for bringing up this important point. We will state and discuss the stationarity assumption and its effects on the interpretation of our results in the revised manuscript. Furthermore, we will discuss seasonal shifts and trends of WT occurrence in more detail in Sect. 3.3. Further details are given in our response to the specific comment on L. 30-31.*

2. The authors used CAP9 as classification but comment that two of the WTs can be considered similar/redundant and that is why CAP7 was preferred by a previous study which is often cited for comparison. It is unclear to me why CAP7 was not preferred over CAP9 provided that throughout the manuscript there are indications that having 9 WTs makes identifiability of WTs more complicated and prone to error.

   *Thank you for this comment. The CAP9 WTs can be understood as the original classification used in operational weather forecasting and climatic analyses by the Swiss Federal Office of Meteorology and Climatology (MeteoSwiss, see Weusthoff, 2011), whereas CAP7 is a simplification made in accordance with the limitations of the methods used by Schwander et al. (2017). Therefore, and also to maintain comparability with future studies using the CAP9 classification by MeteoSwiss, we chose CAP9 as the target of our reconstructions, whereas the CAP7 series was needed to assess the performance in the method inter-comparison. In the revised manuscript we will use CAP7 only in Sect. 3.1 and use CAP9 throughout the rest of our analyses to avoid confusion for the reader (see also reviewer's comment #3).*

3. Evaluation metrics in the summer season are systematically lower, making one doubt if the Weather Type classification is suffering from an under-representation of the atmospheric variable amplitude which is typically low in the summer season and high in the winter season (i.e., PCA input data has not been normalized by the seasonal cycle standard deviation). This aspect is important and should be clarified (see specific comment L.306-307).

   *This is an important point. The calculation of the CAP9 WT classification accounts for*

seasonal differences in pressure variation amplitude (see Weusthoff, 2011). Regarding our reconstructions we assessed several approaches to address this issue. We examined a seasonally dependent standardization of pressure observations (L. 171 f), as well as training individual models for each season (L. 459 ff). Both ideas had to be omitted as no consistent improvements in WT attribution could be determined, but on the contrary results at least partially deteriorated. We will try to emphasize this topic more in the revised manuscript. Further details are given in our response to the specific comment on L. 306-307.

**Specific comments (section addressing individual scientific questions/issues)**

L.12-13 – "In Europe, where daily weather is mainly governed by transient high and low pressure systems driven by the westerly jet stream". It seems a bit simplistic, especially because there are differences between the north-northwest part of the domain, influenced by the Atlantic, and the south-southeast part of the domain, where that influence is smaller and the Mediterranean acts as a frontier between the warm south and cold north.

Thank you for this comment. We agree that our statement omits many processes influencing European weather; we wanted to emphasize the important role of mentioned pressure systems. We will change this (in accordance with our response to the reviewer's comment #3) to „In regions such as Europe, where daily weather is largely governed by transient high and low pressure systems, such classifications […]".

L.30-31 – "in order to study long-term changes in atmospheric circulation patterns and associated surface effects, long-term series of WT are needed". I think this statement is debatable and not justified in the manuscript. There is an assumption behind it: Weather Type classification is an adequate way to analyze long-term changes in atmospheric circulation, and, more importantly, Weather Types are stationary, meaning that the same Weather Types are there in 1800 as well as in the 2000 – in other words, let's hypothesize that a reanalysis existed in 1700, applying a principal component analysis to e.g. geopotential height at 500 hPa to the period 1750-1800 and repeating the same to the period 1950-2000 would yield the same or similar EOFs and in turn describe the same patterns. These two are assumptions on which your paper is based upon, which deserve attention, cannot be taken for granted and should be clearly stated before carrying out your study.

Thank you for this suggestion. We agree that it is important to mention this stationarity assumption when analyzing our findings. The characteristics of typical synoptic situations may have changed over the course of the last centuries and deriving e.g. the CAP9 weather type classification with data from the early 1700s may lead to different weather types than for our reference period in the 20th century. However, for deriving typical WT classifications for periods back in the past by analyzing EOFs of past synoptic patterns and in consequence to detect such changes in the characteristics of governing WTs, the scarce station data available is insufficient. For this reason, our approach of reconstructing a (stationary) set of defined „modern" weather types for the past constitutes the only way possible to extend WT classification that far back in the past and allows to gain important information even despite the limitations that such a stationarity assumption may have. Whereas we cannot exclude slight changes of typical circulation patterns with this approach, the fact that average detection probabilities of reconstructed WTs are high (Fig. 6) throughout the last 300 years points to the validity of the stationarity assumption, as strong changes in the characteristics of governing WTs would lead to a decreased detection probability further back in the past. Also a lack

of trends in the reconstructed WT series can be interpreted as supporting the assumption in the way that changes in the governing WT characteristics over the last centuries are – if at all – small. Furthermore, by analyzing changes of a consistent, „stationary" set of WTs in the past, changes in their occurrence frequency certainly hold important information on past variability or even long term shifts with respect to large-scale atmospheric circulation. In the revised manuscript, we will elaborate on this point in Sect. 2.1 and 3.3 and change the statement in L. 30f to: „By creating long–term time-series of WT classifications, important information may be gained to study long–term changes (i.e. over multiple decades or even centuries) in atmospheric circulation patterns and associated surface effects."

L.40 – When you discuss the limitations of station-based reconstructions you could also mention that weather types generally describe atmospheric circulation over relatively large areas, so going beyond measures from a single point. Also: have these series been detrended?

Thank you for this suggestion. We will include it in the revised manuscript. As described in Sect. 2.2, temperature series have been detrended. Furthermore, all series have been bias-corrected with respect to the monthly EKF400v2 dataset, thus eliminating artificial trends or break points (also described in Sect. 2.2).

L.63 – You wrote that you use CAP7 for the study but then at the end of the section you write that you reconstruct WTs extending the CAP9. Please clarify.

Thanks for this comment. In accordance with our response to your initial comment No. 2, as well as to similar comments in review #3, we will simplify and clarify the use of the CAP7 and CAP9 WT classifications in the revised manuscript, including the line mentioned here.

L.65 – "It does not suffer from subjective WT classes". WT classifications suffer from subjectivity because the choice of the number of classes is subjective unless there is a metric that helps choosing that number (e.g. BIC, Bayesian Inference Criterion, in Falkena et al. 2020).

Thank you for this comment. Whereas the choice of the number of classes in the CAP classification is subjective (see Ekstroem et al., 2002 for details), the term „subjective" in this context refers to WT classification based on expert judgement, i.e. based on visual analyses of hand-drawn weather maps involving personal „subjective" decisions. Those areare distinguished from „objective" classifications (automated, based on statistical approaches using quantitative information) after Philipp et al., (2010). This terminology was introduced in L. 22 ff. We will try to clarify this in the revised manuscript.

L.76-78 – I don't understand the purpose of these lines and perhaps it could be introduced if the authors (or other studies) had assessed the added value of wind direction on periods where this type of record is available.

Thanks for this comment. We refer to the ability of machine learning approaches to include also qualitative information (see L. 50), which could complement quantitative data and – especially in the earlier period – is more often recorded (but not digitized) than quantitative measurements. We will make this clearer in the revised manuscript.

To our knowledge, no other study has yet assessed the added value of wind direction. Neither have we been able to assess this for the reason given in L. 78.

L.81-82 – Why base the study on two classifications CAP7 and CAP9? And not one of the two?

Thank you for this question. We would like to refer to our response to your 2nd introductory remark, as well as your comment to L. 63.

L.103 – How are the WT classified into advective and convective? Please explain.

Thanks for this question. This classification is taken from Weusthoff (2011) which identified the dominating process (advective or convective) in a given WT class. We will indicate this in the revised manuscript.

L.104 – I understand that the WT are computed all year round implying that the larger amplitude of the variations of the atmospheric variables in the winter will potentially bias the WT towards winter patterns. Is there some sort of normalization of this amplitude throughout the year? Please clarify.

Thank you for this comment. Indeed, WTs are computed all year round. As tests with a seasonal normalization (see L. 170 ff), as well as with training individual models for each season (see L. 459 ff.) did not yield any clear benefits, we chose not to include a seasonal dependence, but to train the model on the full set of data. An idea behind this was that machine learning algorithms can eventually also detect seasonal differences in the distribution of input variables and thus correctly attribute WTs for individual seasons. Directly retracing this capability through the numerous connections within a neural network unfortunately is not possible. However, whereas small seasonal differences are apparent in Tables 2 and 3, a consistent bias towards winter patterns cannot be detected in Fig. 7. This supports the previous assumption and our methodical choice not to determine a different treatment for individual seasons.

L.107 – If some of the WT are hard to distinguish from one another as you write, why didn't you use CAP7 also for training your machine learning models?

Thank you for this question. As mentioned in our response to the reviewer's initial comment No. 2, CAP9 represents the „original" WT classification, whereas CAP7 is a simplification of the former made by Schwander et al. (2017), as their method struggled with distinguishing said WTs. We targeted reconstructions to extend the original CAP9 dataset back into the past to ensure comparability with future studies based on this more common (see Weusthoff, 2011) WT classification (and as first tests with ML approaches showed good results with respect to the similar WT pairs). CAP7 was thus merely used for the model intercomparison.

L.191-194 – "Increasing number of covariates can lead to overfitting of the model", I guess this characteristic is valid not only for this method. Also, could you clarify the choice of 4 as threshold for the VIF?

Thank you for this insightful comment. We agree that overfitting can occur in any statistical model with multiple predictors, particularly when linear combinations of predictors (e.g., in logistic regression) or functions (e.g., Generalized Additive Models) are used. In this sentence, we were specifically referring to overfitting caused by multicollinearity. If two predictors are strongly correlated, failing to account for multicollinearity can distort predictor estimates, undermine the statistical significance of features, inflate variances, and increase standard errors. This may render a parameter "useless," contributing to the curse of dimensionality without adding any benefits (e.g., better class separation). Overfitting may also be caused simply by adding too many predictors, even if they are not correlated, leading to good training performance but poor generalization in validation or testing datasets. While this type of overfitting can also happen in machine learning models, we addressed this by using two loops of 10/8 datasets to optimize model performance across all data

splits and did independent testing, effectively limiting overfitting.

Regarding multicollinearity, many machine learning algorithms are more robust. Algorithms that internally perform feature selection or use regularization techniques (as done in all the ML models tested in our study) are generally less vulnerable to multicollinearity due to their non-parametric nature.

To make this clearer we will change L. 191-194 accordingly.

On the choice of a VIF threshold of 4, we selected this as it is a commonly accepted conservative threshold in statistical modeling. In our testing, higher thresholds (e.g., 5 or 10) led to diminished performance in the validation datasets.

L.221-222- This characteristic is crucial: "As circulation patterns can persist several days", as WT must persist a few days on average. The average persistence in days of each CAP7 (or CAP9) should be added to the manuscript along with a contingency table with weather patterns in rows and columns with shares (or counts) of transitions (e.g., see table 1 / 2 in Robertson et al. 2020). I wonder if transitions and preferential paths of transitions among WTs should be fed to the different machine learning approaches. Please comment.

Thank you; persistence and preferential transition paths of WTs are both important points, which we shall include in the supplement to the revised manuscript in the form of the suggested contingency tables. Regarding the suggestion of feeding transitions / preferential paths to the ML models: as both time-dependent models (RNN / CNN, taking into account information from three consecutive days) did not yield consistent benefits compared to feedforward NNs, adding time-dependent information such as probabilities or preferential paths of transitions to the model input (in addition to the fact that the WTs of the previous day would have to be known) is unlikely to improve model performance.

L.243-245 – As noted above, choice of CAP7 over CAP9. To make things comparable with Schwander et al. 2017 things are adjusted between the two classifications (cap7, cap9) in a way that it seems like a single one would have been more convenient.

Thanks for this suggestion. We will simplify and clarify the use of CAP7 and CAP9 in the revised manuscript in order to avoid confusion (see our response to the initial comment No. 2, as well as to suggestons in the reviewer's comment #3).

L.267-273 – In light of the error in the model set up found in Schwander et al. 2017 I wonder if it is worth following their footsteps so closely. I acknowledge the importance of having a reference study to compare to, but perhaps the authors could have been more brave in overcoming that study.

Thank you for this comment. Whereas the error in the model set up found in Schwander et al. (2017) represents a strong limitation of their approach, we feel that it serves as an excellent baseline also as the operational attribution of WTs (see e.g. Weusthoff, 2011) follows similar approaches. In the revised manuscript, we will, however, move comparisons with CAP7 in Sect. 3.3 to the supplement in order to follow a clearer structure and enhance our independence of that study.

L.296 – Table 2 shows that, of the four methods proposed, it seems like NN and RNN outperform the other methods, with RF always behind regardless of the number of stations.

Thank you for this observation. Yes, RF shows slightly lower accuracies (by 2 – 3 %) than the neural networks. We will mention this explicitly in the revised manuscript.

L.303-305 – I think this statement is not sufficiently supported. Are there works that have carried out similar analysis with statistical approaches that do not involve machine learning?

Thank you for this comment. To our knowledge, the studies by Schwander et al. (2017) and Delaygue et al. (2019) provide the only station-based European WT reconstructions and do not involve machine learning; both studies, however, do not go into detail about such methodological limitations. Nevertheless, a distance measure (even a statistical distance) is more rigid with respect to complex, non-linear patterns than machine learning approaches. We thus think that from a theoretical point of view, this statement can be considered valid. We will specify „capture details in the data and non–linear effects" to „better fit non-linear relationships and interactions in the data" and emphasize the hypothetical nature of the statement in the revised manuscript.

L.306-307 – Well, as noted above, this does not come as a surprise if no normalization of the atmospheric field was carried out prior to the computation of the regimes (the standard deviation during the year varies considerably with very low in the summer compared to the winter months e.g. figure 1d in Lee et al. 2023). This aspect is important and should be clarified.

Thank you for this comment. We agree that the seasonal differences in pressure variations should receive a more prominent position, which we will introduce in Sect. 2.2 in the revised manuscript. Whereas these differences have been accounted for in the CAP9 WT classification (Weusthoff, 2011), tests with seasonally dependent standardization of pressure observations (L. 171 f), as well as seasonal model training (L. 459 ff) unfortunately did not yield better results compared to using the full set of raw pressure observations. However, as seasonal variation of accuracy and HSSs in Tables 2 and 3 (especially the latter) are small compared to the baseline approach, ML methods appear to be able to tackle this issue at least to a certain extent, as argued in L. 306 ff.

L.309 – In light of the drops in accuracy in the summer this statement is perhaps optimistic and limited to the comparison to Schwander et al. 2017. "Our models are better capable of coping with seasonal differences".

Thank you. We will change this sentence to „[…] models are better capable of coping with seasonal differences although some seasonal patterns in the accuracy remain." to put the statement in a clearer context.

L.350 – increased accuracy in fall and winter, otherwise for summer months – another clue in the direction of lacking summer information (normalization of atmospheric field)? Or the fact that wet days are more frequent in fall and winter as opposed to summer months (regardless of the type of precipitation – large scale in winter vs. convective in summer as noted at L. 355)?

Thank you for these considerations. The evaluation for the station sets of 1728 and 1864 did not show a uniform pattern with worse performance for summer months and better performance for winter months. We therefore cannot make such a conclusion regarding the use of wet days.

L.367 – I struggle with the term "accuracy" which, in e.g. operational forecasts, relies on the evaluation of simulated vs. actual variable values. In this case the actual values cannot be used as they can only be reconstructed. Therefore, is it appropriate to use this term? I suggest the authors clarify this point at the beginning of the paper, either in the Introduction or in the Data and Methods sections.

Thank you for this comment. The accuracy in this case refers to the station sets for the historical period with respect to a modern reference period 1957–2020. In accordance with a similar comment

made by reviewer #2, we will change the phrasing in L. 367 ff in order to be transparent about this issue: „The achieved accuracy using the smallest station set (stations available from 01.01.1728 to 31.12.1737) is already remarkably high […]. Adding more stations […]. Note that validation metrics shown in Table 3 only provide values with respect to the reference period 1957-2020. The actual values for the past periods may be lower due to larger uncertainties and errors in the data, but unfortunately cannot be determined due to the lack of a historical reference WT series."

L.370 – Summer months lower accuracies AND L. 399 – False WT predictions in summer seems to originate from other sources - related to year-round WT classification?

Thank you for this remark. The year-round WT classification (i.e. indifference to the seasonality of pressure variance) is certainly linked to the results shown in Table 3 (L. 370), as well as in Fig. 5b (L. 399), which is stated in L. 400. As for other passages in the manuscript, we will clarify this link in the revised version.

L.402 – "Weather types might change over the course of one day". Are you sure this characteristic is relevant in errors assigning WTs? Aren't WTs on average lasting 2+ days?

Thanks for this question. The daily CAP9 classification as described in Weusthoff (2011) tries to attribute instantaneous (i.e. daily) synoptic situations rather than patterns persisting over several days (see e.g. Mittermeier et al., 2022). Consequently, CAP9 does not include an undefined/transitional WT category. Whereas some WTs (e.g. WT 8) on average persist for two or more days, WTs occurring only for one day make up approximately one third of all situations and are prone to false predictions (as indicated in L. 405 f). As such changing conditions leave mixed imprints on daily averages of measured variables, the issue seems relevant for the reconstruction of daily WTs.

L.408 – Please clarify what you mean by "transient WTs".

Thanks for this remark. We'll change „transient WTs" to „transient situations" (see response to RC #3), for which a definition is given in L. 406.

L.415 – It is of great value that the NN attributes a probability of occurrence to all WTs, and I think this feature should be discussed further in the assessment of the good/bad WT daily classifications. I would expect that the WTs with highest probability isn't always with values of 0.8 or above and that days in which probabilities are more evenly distributed among the 9 classes exist. E.g. WT1 0.1, WT2 0.1, WT3 0.1, WT4 0.1, WT5 0.1, WT6 0.1, WT7 0.1, WT8 0.14, WT9 0.16, In this case what is the chosen WT, the WT9? One could argue that "no regime" class would be a more suitable choice. Have you counted how many times the probability of the winning WT is not crystal clear (probability much larger than the remaining WTs)?

Thank you for this comment. In Fig. 6b, we show the probability of the winning WT with respect to correct and false classifications. Whereas the model is confident for the correctly assigned WT (probabilities > 0.7), for false detections probabilities as low as 0.35 are apparent. Although not as extreme as the example given by the reviewer (in which case – as a side note – WT 9 would be chosen), the probability of winning categories especially for wrongly assigned WTs is not always crystal clear. As the CAP9 series does not have a transitional / neutral „no regime" class, we did not want to introduce such a class for our reconstructions, even though it might be a suitable choice. However, as probabilities are provided together with the respective WTs in the published series, users may choose to introduce such a class for uncertain cases.

L.431 /Figure 7 – Biases are visibly low for WT 8 and WT 9, do the authors have an explanation for this? From Figure 9 it seems that these two occur very little in the summer.

Thanks for this question. First, the small biases are directly linked to the low occurrence frequency, as in Fig. 7, the percentage of the biases is shown with respect to the number of days in a year. We will make this clearer in the revised manuscript. Second, WTs 8 and 9 show among the highest accuracies (see Fig. 3) for the individual WTs. Low bias values thus are to be expected.

L.450 – On the absence of artificial discontinuities: it makes no sense to comment on discontinuities using the eye over a plot with smoothed lines (10yrs running mean). Why don't the authors apply a statistical test for discontinuities/change-points on the non-smoothed series?

Thank you for this excellent suggestion. We will include a statistical test for discontinuities / change-points applied on the reconstructed WT series in order to support our statement in the revised manuscript.

L.457 – "artificial trends can be dectected" – have you found significant trends through the application of a statistical test? It would be interesting to know if/which WTs have become more or less frequent throughout the period of analysis and if WTs occurrences have shifted in season.

Thanks for mentioning this important point. We tested the yearly WT occurrence, as well as 10-year running averages for linear trends (linear regression with t-test) over the full 300 year reconstructions. Considering $\alpha = 0.05$, no significant trends could be detected for any WT. We did, however, find significant trends for individual seasons. An examination of our new reconstructions with respect to trends in WT occurrence is still ongoing and was thus not included in the presented manuscript. We will include a sentence in the revised manuscript, that statistical tests have been applied to detect trends.

L.470 – use either "thus" or "indeed".

Thanks. We will opt for „indeed" in the revised manuscript.

L.474-475 – I found no description of the detection method for trends and discontinuities in the manuscript.

Thank you for this remark. As mentioned in our response to the comments on L. 450 and L. 457, we will add results from our trend analysis and a yet-to-implement break-point detection method applied to the reconstructed WT series in the revised manuscript.

L.484 – "WTs with low occurrence and strong seasonality can pose a challenge for reconstructing WTs", this is why I wonder why CAP9 was preferred over CAP7 (fewer WTs).

Thank you for this comment. We would like to refer to our response to your initial comment No. 2 which treats this issue.

L.488-490 – "Transient WTs make the distinction on a daily resolution difficult,… issue might be solved with the use of subdaily data". I consider this option inadequate for the very nature of reconstructing WTs back to 1700s, it is already a miracle if you get a daily value, imagine subdaily, utter wishful thinking! Also, as far as transient WTs are there and may hinder daily classification, the degree to which the knowledge of sub-daily WTs would help such classification is far from demonstrated. WTs are, by design, approximation of reality at at daily time scale, it is to be expected that in some days a good match with the archetypal WT is lacking, it's part of the game.

Thank you for this remark. First, we would like to emphasize that subdaily station records reaching back to the 1800s and even 1700s are by far not as much wishful thinking as the reviewer suggests. Efforts to gather historical meteorological data such as the International Surface Pressure Databank (ISPD; Compo et al., 2019) have made available multiple sub-daily pressure time series back to the early 19$^{th}$ century. Also, some long and homogenized sub-daily series reaching back even to the 17$^{th}$ century have been recently introduced (Cornes et al., 2023). Nevertheless, more digitization and homogenization efforts focusing on sub-daily data would be needed to provide a robust basis for WT reconstruction. We totally agree with the reviewer that synoptic situations sometimes do not match well with archetypal WTs; the issue of transient WTs could nevertheless be improved by using sub-daily information.

**References:**

Falkena SK, de Wiljes J, Weisheimer A, Shepherd TG. Revisiting the identification of wintertime atmospheric circulation regimes in the Euro-Atlantic sector. Q J R Meteorol Soc. 2020; 146: 2801–2814. https://doi.org/10.1002/qj.3818

Robertson, A. W., N. Vigaud, J. Yuan, and M. K. Tippett, 2020: Toward Identifying Subseasonal Forecasts of Opportunity Using North American Weather Regimes. Mon. Wea. Rev., 148, 1861–1875, https://doi.org/10.1175/MWR-D-19-0285.1.

Lee, S. H., M. K. Tippett, and L. M. Polvani, 2023: A New Year-Round Weather Regime Classification for North America. J. Climate, 36, 7091–7108, https://doi.org/10.1175/JCLI-D-23-0214.1.

---

## Author Comment (AC3)

**RC3: Review of „Weather Type Reconstruction using Machine Learning Approaches" by Pfister et al.**

**Summary**

The authors assess various machine learning approaches for weather type reconstruction and then use the best-performing model to extend the time series of weather types over central Europe back to 1728. I consider the study a worthwhile addition to the effort of reconstructing past weather, but the authors should first address some questions that I raise below.

We'd like to thank the anonymous reviewer for his thorough assessment and his valuable comments to improve our manuscript.

**Major comments**

The authors try to make their results comparable with the study by Schwander et al. (2017), which has a marked negative impact on the readability of the text (e.g., in 81, 108, 150, 247, 268, etc.). Repeated efforts to explain differences in the dataset and methods, especially the constant switching between the CAP7 and CAP9 methods, lead to unnecessary confusion for the reader. Additionally, if I understand correctly, different classifications were used in validation and reconstruction, which could be considered questionable, to put it mildly. I suggest moving the discussion/comparison of the authors' results with the other study to one paragraph in Section 3, including Schwander's method among the methods they validate, and consistently using either the CAP9 or the CAP7 method. Schwander's model was already recalculated by the authors (and an error in the original calculation was found), which means that the change will be relatively easy.

Thank you for this suggestion. We agree that switching between the two closely related weather type (WT) classifications (CAP7 and CAP9) throughout the paper might lead to a certain amount of confusion for the reader, as was also pointed out in the reviewer's comment #1. We will therefore simplify the use of the two different WT classifications in the revised manuscript and restrict analyses using CAP7 to Sect. 3.1, while CAP9 is used in the rest of the manuscript.

As you remarked correctly, in Sect. 3.1, the machine learning approaches were trained on the CAP9 WTs, whereas the baseline approach (Schwander et al., 2017) was calculated for the CAP7 WT classification. Results of a validation with CAP9 (ML approaches only, not shown) revealed similar patterns to the validation with CAP7. In order to be able to compare all approaches, we show the validation results (Table 2) only for the CAP7 weather types, meaning that the 9 weather types predicted by the machine learning approaches are reduced to 7 WTs accordingly. We deem this comparison and the accompanied difference of the WT classification series used for model training and validation suitable for the following reasons:
- the CAP7 classification has been derived from the original CAP9 WT classification and the weather types of both classifications exactly correspond to each other with the exception that WT pairs 5 and 8, as well as 7 and 9 are merged. Implications with respect to training on CAP7 vs. CAP9 can thus be considered minor (i.e. a slight underestimation of accuracy).
- Our aim was to accurately reconstruct the CAP9 weather type classification using machine learning which is among other applications used in operational weather forecasting and climatological analyses (Weusthoff, 2011). The reduced CAP7 classification has been introduced by Schwander et al. (2017) as they found 9 WTs hard to distinguish with their approach. Using CAP7 would possibly omit relevant information on circulation types.
- As shown in Schwander et al. (2017), the baseline approach is not suitable to predict the 9 weather types of CAP9. Recalculating this approach for CAP9 would thus exceed its already known limitations and thus not be meaningful for a comparison. Therefore, we used CAP7 for model comparison in Sect. 3.1.

In the revised manuscript, we will limit the comparison with CAP7 to Sect. 3.1 and move all such comparisons from Sect. 3.3 to the supplement, as we feel that this information might be interesting for readers working with the CAP7 reconstructions. The main part of the article will then consistently use the original CAP9 weather types.

In Conclusions, the authors should mention that best performing method differs depending on season, used data, and validation metric, instead of the over-generalizing "The feedforward neural network slightly outperformed the other ML approaches..." and that other method(s) lead to comparable results in a shorter time. I would also consider worth stating here that the ML approaches may be less sensitive to the quality of the classification (see my suggestion bellow, l. 306).

Thank you for this comment. We will specify this in more detail in the conclusions following your suggestion.

**Minor comments**

„in the accuracy of the used methods" is rather vague
Thank you. Indeed, the wording is not really precise about the nature of the restrictions of WT reconstructions. We will change this to „by methodical limitations".

CAP9 abbreviation is used but not defined
Thank you. We will introduce the abbreviation in the revised abstract.

WT abbreviation is defined here but not used consistently in the paper
Thank you for noticing. We will harmonize its use in the revised manuscript.

Can you refer to a global-scale evaluation of the skill of classifications that would confirm your claim that „In Europe classifications prove particularly useful to describe the prevailing atmospheric conditions?"
Thank you for this question. Our statement here seems to have missed its point. Our aim was to state that particularly in regions where weather is largely governed by large-scale circulation (such as Europe), WT classifications are particularly useful, rather than emphasize the role of Europe with respect to other regions in the world. We'll rephrase this in the revised manuscript (see also our response to reviewer's comment #1): „In regions such as Europe, where daily weather is largely governed by transient high and low pressure systems, such classifications […]"

Given your references here, probably "model outputs" would be more appropriate and clearer than "weather forecasting model simulations"
Thanks for this suggestion. As it is important to state the type of model, we will opt for „weather forecast model outputs".

"introduced, that" –> "introduced that"
Thank you for hinting at this. We will adjust in the revised manuscript.

"With the newest generation of reanalysis datasets, many WT records could already be extended back to the 19th century (Philipp et al., 2010; Jones et al., 2014)." How do these two rather old references support your claim regarding the newest reanalyses?

Thank you for this observation. Whereas the newest generation of reanalyses (i.e. particularly 20CRv3) would allow to extend WT reconstructions as far back as the early 19[th] century, the cited papers used older versions of reanalysis datasets. We will adjust the phrasing in the revised manuscript accordingly.

Maybe "machine learning" would be better than „artificial intelligence"?
We absolutely agree with this suggestion; thank you.

50-55 First, you write that "artificial intelligence is commonly used for classification...in climatological research" and then "In the context of WT classifications, ML is still a rather novel approach." Please clarify. Furthermore, cluster analysis is commonly accepted as a machine learning approach and some clustering methods are among the oldest WT classification methods. Therefore, I would oppose your latter claim.
Thank you for these comments.
Regarding your first point: whereas artificial intelligence is used for classification and pattern recognition tasks related to the examples given in L. 50-55, we found only few applications of AI to WT classification specifically. We'll try to make this clearer in the revised manuscript.
Regarding the second point: the distinction between what are common statistical approaches and what is machine learning may be an issue of discussion. ML in this context was meant to refer to more advanced approaches like random forests or neural networks. Clustering approaches certainly are among the oldest and most prominent approaches for WT classification, but we would not necessarily count them as machine learning in this context.

It is not clear what "this pioneering work" refers to
Thank you. We refer to the three aforementioned references which used modern ML approaches for WT reconstruction. We will try to make this clearer in the revised manuscript.

Please add a little bit more detail to the decomposition; for example, what similarity matrix is decomposed?
Thank you for this excellent suggestion. We will include more details on the PCA step in the revised manuscript.

I'd suggest "the first step... the second step" rather than "a first...a second"
Thanks. We'll change the wording accordingly.

Fig. 1 Some stations in the figure (e.g., Cadiz) are not visible. I suggest moving the symbols in front of the coastal lines and increasing the size a little
Thank you for this suggestion. We'll adjust the figure accordingly in the revised manuscript.

ERA5 goes back to 1940, why was only 1957–2020 used?
Thank you for this question. We chose the shorter period as it corresponds to our reference period for which the CAP9 WT series is available.

Instead of "are well distributed across most parts of Europe" I would suggest a considerably more accurate "are relatively well distributed across central Europe", or similar.
Thanks for this suggestion. We'll state the geographical distribution more precisely in the revised manuscript.

Probably "SLP" instead of "pressure" would be clearer and more accurate. This occurs several times throughout the paper
Thank you. We will change this to „sea level pressure" in the revised manuscript.

It is not clear what "the latter study" refers to
Thanks. It should refer to Schwander et al. (2017). We will adjust the phrasing to make this clear.

Is not "The station sets used for the reconstruction of CAP9 WTs (Sect. 3.3.1) are summarised in Fig. 2" at odds with "Figure 2. Station sets of a) pressure and b) temperature used for the model comparison", considering that CAP7 was used for comparison?
Thank you for this detailed observation. Indeed, Fig. 2 shows station sets used for both, comparison and reconstruction. We will adjust text and figure caption accordingly.

"Thus, temperature data were corrected for their seasonality by fitting the first two harmonics to each temperature record, which was then subtracted from the data." should be made clearer; your sentence suggests that the temperature record was removed. I suggest "Thus, temperature data were corrected for seasonality by fitting the first two harmonics to each temperature record and then subtracting these harmonics from the data."
Thanks for this excellent suggestion. We'll adopt the suggested phrasing in the revised manuscript.

"their contribution" is not clear
Thank you for mentioning this point. We will rephrase it in the revised manuscript: „[…] trend or seasonality, which contribute only a negligible part to the total variability of these variables.

"they are trained" is not clear
Thank you. We will change this to „the former".

"trained on the same station data" is surprising to me, since above you explained considerable differences between your dataset and that used by Schwander et al.
Thank you. In fact, we state in L. 120 ff. that the station data used for the model comparison is the same as in Schwander et al. (2017). The described differences refer to additional station series used for our CAP9 reconstructions presented in Sect. 3.2 and 3.3.

"both, hyperparameter" –> "both hyperparameter"
Thanks.

Table 2 Is it necessary to repeat "Acc =" and "HSS=" for all values? Did you consider another type of graphical output, which may be more readable?
Thank you for your suggestion. Given the large amount of combinations of station sets and models and the sometimes very small differences between the validation metrics, we deemed it more suitable to present the detailed quantitative results in a table instead of a graphical output. To make Tables 2 and 3 better readable in the revised manuscript, we will use different font styles for the two validation metrics instead of writing „Acc =" and „HSS =" and add a corresponding explanation in the table captions.

"This best–performing model" is not clear
Thank you. We'll change this to „The best-performing MLG model [...]".

306-309 A useful interpretation could be that ML models may be less sensitive to the used classification compared with the simple baseline model. One may argue that projecting summer circulation on 7 WTs trained for an annual time series is far from ideal, because the classification lacks the necessary detail to explain the relatively weak and specific summer circulation. One reason for this is the dominant WT1: if a stand-alone classification was trained by cluster analysis only for summer days, it would find relevant summer patterns that would be more or less equally populated instead of the snowballed WT1.

Thank you for this remark. We agree with your interpretation that ML models are less sensitive to the used classification and especially to seasonal differences compared to our baseline model. However, CAP9 were originally determined from seasonality-corrected pressure data (see Weusthoff, 2011). The dominance of WT1 therein may thus not be interpreted as an inability to capture the specific characteristics of summer circulation due to the definition from annual time series. Regarding the training of our reconstruction models on annual time series, we examined a seasonally dependent standardization (see L. 170 ff), as well as training the models for each season (L. 459 ff), both of which did not yield satisfactory results.

"the overall atmospheric signal seen in a combination of information" is rather vague and unclear
Thank you for this comment. We will change this to „the information on an atmospheric state over a larger region".

I would also consider adding "western Europe" because (south)westerly advection is an important feature of circulation over central Europe
Thank you for this suggestion. Whereas we do have the stations of London, Paris and Cadiz in western Europe, we agree that due to the important role of (south)westerly advection, more stations in this region may yield benefits. We will add this region to the list in the revised manuscript.

I think that "The model comparison revealed the feedforward neural network (NN) to exhibit the highest accuracy and HSS estimates" is an exaggeration and at least "on average" should be added into the main clause. However, to me it seems that the NN, RNN and CNN lead to nearly identical results and Table 2 shows that the best performing method is sensitive to the choice of season, data and validation metric.
Thank you for this point. We will rephrase this sentence in the revised manuscript.

377-9 "For a true WT 8 most false predictions show WT 5, and for WT 9 most false predictions show WT 7. Already Schwander et al. (2017) found these two pairs hard to distinguish, leading them to reduce the number of WTs accordingly." I do not think that the results shown in Table 3 support Schwander's claim. First, accuracy for WTs 8 and 9 are not worse than that for the other WTs. Second, these two WTs are outliers – if you imagine the 9 WTs ordered such that their position (for instance in a 1D or 2D Sammon map, the first PC plain, etc.) respects their position in the high-dimensional space, the two outlying WTs would simply neighbour with fewer WTs. Consequently, most (all) of the false hits will be linked to the one "closest" WT (which has a strongly correlated but weaker circulation pattern). I would not even consider this an artefact of the methodology but rather a geometric necessity.
Thank you for this remark. We absolutely agree that false predictions for the extreme WTs 8 and 9 will most likely be attributed to the „closest" WTs, in this case WTs 5 and 7, independent of the methodology, and as you correctly state, our accuracies for these extreme WTs are not lower than for other WTs (whereas Schwander et al. (2017) seem to have had large difficulties to distinguish between the extreme and „closest" WTs). We did not mean to support Schwander's claim with our interpretation of Fig. 3, but to emphasize the „false detection pattern" for the extreme WTs and the general capability of ML models to correctly predict them. We will change this section to:
„For the „extreme" WTs 8 and 9, most false predictions – as expected – identified WTs 5 and 7, which show the most similar patterns to the correct WTs 8 and 9, respectively (compare Fig. 1). Whereas Schwander et al. (2017) found these two WT pairs hard to distinguish and reduced the number of WTs accordingly, the NN model accuracies for WTs 8 and 9 are comparable to the other WTs. The NN model is thus capable of correctly distinguishing between these „extreme" (i.e. with respect to the intensity and extent of high/low pressure systems) WTs and their less extreme counterparts."

Figure 3: Consider specifying what are the "reference period" and "reference CAP9 series" in this case. In the validation phase, it is clear what accuracy means. However, this is not the case for the cross-validation periods

Thanks for this suggestion. We will include such an explanation in L. 375 in the revised manuscript.

Paragraph starting at 380+Figure 4 This could use some additional introduction and explanation – you lost me here

Thank you for this comment. We will add the following explanation in L. 384 in order to help the reader understand Fig. 4: „Deviations of the red and blue circles at individual/all observation points indicate regional/overall discrepancies in the observed pressure distribution as reason for false detection. Coinciding red and blue circles would mean that observation patterns of true and false predictions are identical and that the reason for false predictions are not explainable from the observations."

Figure 5 Showing true and false maps as deviations from obs would probably support your interpretation more clearly

Thank you for this suggestion. We originally considered showing deviation maps with respect to observations. However, this blurs information on the position of low and high pressure systems in the false prediction maps. For the reader, it would be harder to see whether wrongly predicted patterns are just weaker (e.g. WT8 in Fig. 5a) or whether inverse pressure systems are apparent in certain regions (WT9).

408+409+488 Consider "transient situations" or similar instead of "transient WTs". Transient WTs are WTs with low persistence, but you did not show that.

Thanks for this suggestion. We'll adopt it in our revised manuscript.

"The chosen WT for these cases might be arbitrary depending on slightly stronger patterns (i.e. dominating by a small margin)" Please reword, I do not follow

We'll rephrase this sentence in the revised manuscript: „The chosen WT for these cases typically is the one with the strongest imprint on the daily average station observations and not necessarily the one persisting throughout most of the day. Furthermore, a dominating WT might be chosen by a very small margin."

"or by calculating WTs for a specific time of the day" I do not believe that this would have any effect on the presence of transient/boundary circulation fields in your data. Circulation fields form a continuum of patterns and I do not know of any reason to expect that instantaneous and averaged fields differ in this respect. Ditto 489

Thank you for this comment. Whereas daily averages of station observations may have blurred/mixed information from two or even three WTs in a transient situation, measurements for a specific time (e.g. 12 UTC) would provide a sharper pattern more likely to be attributable to one specific weather type. Hence, we would certainly expect some improvement with respect to using daily observations.

Figure 6b boxplots add y-axis labels (%?) and explanation of shown percentiles

Thanks for this suggestion. Figure 6b shares its y-axis with Fig. 6a; we'll adjust the figure to make this clearer in the revised manuscript and add an explanation of shown percentiles in the figure captions.

Figure 8 and especially 9: Consider using an even longer filter, the lines are still very noisy

Thank you for this suggestion. Our aim was to show the large year-to-year variability of WT occurrence. However, to make the figures better readable, we will remove CAP7 in the revised manuscript (see response to RC1).

"The results presented in Fig. 8 suggest that the reconstructed CAP9 time series do not show any apparent artificial discontinuities that go beyond natural variability." How was this tested? You present this in Conclusions as one the main results but you did not provide any testing

Thank you for this important point. In accordance with the reviewer's comment #1 we will include results from statistical tests with respect to trends and discontinuities in the revised manuscript.

"Our results emphasize the importance of constantly improving WT classification methods…" One may argue that you did not test or enhance the CAP9 methodology itself, therefore rewording the sentence may be advised

Thank you for this suggestion. We'll change this to „Our results emphasise the importance of continuously improving methods of WT reconstruction..."

"occurrences" or similar may be better than "samples"

Thank you. We'll change „number of samples" to „occurrence frequency" in the revised manuscript.

"Adding seasons as additional predictors or training different models per season could solve this issue, although the sample size of rare WTs might be too small." If season-specific classifications are trained (and a suitable classification method utilized that does not tend to identify marginal WTs), the issue of WTs with marginal occurrences will disappear. Additionally, season-specific classification could have fewer WTs, which would (I suppose) decrease the computation cost

Thank you for this comment. Whereas season-specific classifications (with an aptly trained classification model, of course) may considerably improve the issue of WTs with a strong seasonality or marginal occurrence, we wanted to point out some crucial issues related to that: smaller training datasets typically deteriorate the robustness of machine learning models. Especially a smaller number of samples of marginal WTs leads to an under- or overrepresentation of these WTs in the training dataset, exacerbating model training and possibly leading to an under- or overrepresentation of marginal WTs in the reconstructions. Furthermore, „hard-coding" seasons into the reconstruction model like this may ignore seasonal shifts in WT occurrence. Regarding computational costs, training multiple models (i.e. for each season) with smaller input datasets does not hold much benefits with respect to training one model on a larger dataset.

The availability of monthly gridded datasets is mentioned in the paper. I was wondering whether including monthly SLP patterns as one the predictors could improve the models.

Thank you for this suggestion. Despite the idea being attractive for reasons of data availability, using monthly SLP averages directly as a predictor for the ML models is unlikely to improve results, as an artificial tendency towards an „average monthly WT", as well as discontinuities in WT occurrence at the turn of each month would be introduced.

I would also welcome a note on the possible sensitivity of the models to the chosen classification and its parameters. Classification method is one of the major factors of any synoptic-climatological study and I would expect a significant sensitivity of WT reconstructions (and validation metrics) to changes in the classification methodology.

Thank you for this remark. Our article focuses on a single pre-defined weather type classification (CAP9) in order to compare the skills of different machine learning approaches to a baseline approach provided by Schwander et al. (2017), providing information on the sensitivity of reconstructions with respect to different models and input data (station sets) for this WT classification (Tables 2 and 3). While model performance is certainly sensitive to the chosen WT classification (e.g. dependent on the number of WT classes and their relation to the available station observations used as input, see e.g. the validation in Mittermeier et al., 2022), we did not perform a sensitivity analysis for the ML approaches with respect to other WT classifications (e.g. GWT or

Lamb weather types), as this – although certainly interesting – would go beyond the scope of this paper and thus has to be left for future research.